# WRITE AND PAINT: GENERATIVE VISION-LANGUAGE MODELS ARE UNIFIED MODAL LEARNERS

**Shizhe Diao**[*]
The Hong Kong University of Science and Technology
sdiaoaa@connect.ust.hk

**Wangchunshu Zhou**
ByteDance AI Lab
wangchunshu.zhou@inf.ethz.ch

**Xinsong Zhang**[†]
ByteDance AI Lab
zhangxinsong.0320@bytedance.com

**Jiawei Wang**
Shanghai Jiao Tong University
wjw_sjt@sjtu.edu.cn

## ABSTRACT

Recent advances in vision-language pre-training have pushed the state-of-the-art on various vision-language tasks, making machines more capable of multi-modal writing (image-to-text generation) and painting (text-to-image generation). However, few studies investigate if these two essential capabilities can be learned together and boost each other, making a versatile and powerful multi-modal foundation model. In this work, we disclose the potential of symmetric generative vision-language pre-training in learning to write and paint concurrently, and propose a new unified modal model, named DaVINCI, trained with prefix language modeling and prefix image modeling, a simple generative self-supervised objective on image-text pairs. Thanks to the proposed prefix multi-modal modeling framework, DaVINCI is simple to train, scalable to huge data, adaptable to both writing and painting tasks, and also strong on other vision, text, and multi-modal understanding tasks. DaVINCI achieves competitive performance on a wide range of 27 generation/understanding tasks and demonstrates the superiority of combining vision/language generative pre-training. Furthermore, we carefully benchmark the performance of different vision-language pre-training objectives on different scales of pre-training datasets on a heterogeneous and broad distribution coverage. Our results demonstrate the potential of exploiting self-supervision in both language and vision inputs, and establish new, stronger baselines for future comparisons at different data scales.[1]

## 1 INTRODUCTION

Self-supervised language model pre-training (Peters et al., 2018; Radford et al., 2018; Devlin et al., 2019; Liu et al., 2019; Lewis et al., 2020; Raffel et al., 2020; Brown et al., 2020; Fu et al., 2022; Zhou et al., 2021b; Diao et al., 2020; 2021; Zhou et al., 2021a; Xu et al., 2020; Zhou et al., 2020; 2022a; Pan et al., 2022; Diao et al., 2023) has reshaped the landscape of modern natural language processing (NLP) research, pushing the state-of-the-art of a wide range of NLP tasks. Recently, this success has been transferred to the multi-modal context and resulted in a number of vision-language pre-trained models (VLMs) (Lu et al., 2019; Tan & Bansal, 2019a), achieving state-of-the-art results on various vision-language tasks. Most existing VLMs are BERT-like Transformer (Vaswani et al., 2017) encoders pre-trained with a combination of different vision-language pre-training (VLP) objectives: masked multi-modal modeling (Lu et al., 2019; Tan & Bansal, 2019b; Chen et al., 2020; Li et al., 2020), multi-modal alignment prediction (Lu et al., 2019; Tan & Bansal, 2019b; Chen et al., 2020; Li et al., 2020), region of interest feature regression (Tan & Bansal, 2019b), image-text matching (Li et al., 2021; Zeng et al., 2021), to name a few. However, the roadmap towards large language models reveals a transition pattern from encoder-only models like BERT (Devlin et al., 2019) / RoBERTa (Liu et al., 2019) to sequence-to-sequence models like T5 (Raffel et al., 2020) / BART (Lewis et al., 2020) and autoregressive models like GPT-3 (Brown et al., 2020) / PaLM (Chowdhery et al., 2022) to tackle

---

[*]Work done during the internship at ByteDance AI Lab.

[†]Corresponding author

[1]The code and pre-trained models are available at https://github.com/shizhediao/DaVinci.

more tasks in a unified way, and from complicated objectives like masked language modeling / next sentence prediction / replace token detection to a simple language modeling objective to improve the scalability of pre-training. This suggests that the generative pre-training paradigm with simple targets shows great potential for pre-training more scalable and general VLMs.

To this end, several recent studies (Cho et al., 2021; Zhang et al., 2021a; Wang et al., 2021b; 2022) investigated sequence-to-sequence (seq2seq) vision-language pre-training and achieved state-of-the-art results on a range of vision-language understanding and generation tasks. For example, VL-T5 (Cho et al., 2021), OFA (Wang et al., 2022) and PaLI (Chen et al., 2022) formulate various vision-and-language problems into seq2seq tasks and pre-train a seq2seq VLM by multi-tasking on these tasks. In addition, ERNIE-ViLG (Zhang et al., 2021a) and SimVLM (Wang et al., 2021b) pre-train seq2seq VLMs with a simple language modeling or prefix language modeling objective on a large number of image-caption pairs. While achieving promising results, these objectives are not versatile enough, resulting in VLMs that are only capable of a subset of tasks in image-text modalities. On the other hand, the recent success of generative language pre-training (Brown et al., 2020) and generative vision pre-training (He et al., 2022; Bao et al., 2021) motivates us to explore generative vision-language pre-training to learn more versatile and scalable vision-language models.

In this work, we introduce prefix multi-modal modeling, a unified generative pre-training framework that extends prefix language modeling to the multi-modal context and learns a multi-modal foundation model by learning to write and paint simultaneously. As illustrated in Figure 1, given an image-caption pair, we split the image and caption into two parts denoted as prefix and suffix. To make prefix image modeling compatible with the seq2seq formulation of conventional prefix language modeling, we follow DALLE (Ramesh et al., 2021) and convert images into discrete sequences of image tokens (van den Oord et al., 2017). We then train the model to generate the suffix in one modality based on the prefix in the same modality and the complete input in the other modality. In this way, prefix multi-modal modeling can fully exploit self-supervision from large-scale image-caption pairs by learning to write and paint simultaneously. We pre-train DAVINCI [2], a vision-language foundation model, with the proposed prefix multi-modal modeling framework on large-scale image-text pairs. DAVINCI is the first self-supervised vision-language foundation model that is versatile for all kinds of tasks in vision-and-language modalities, including image-to-text generation, text-to-image generation, vision-language understanding, and single-modal language / vision tasks. DAVINCI consistently outperforms FLAVA (Singh et al., 2021), an existing vision-language foundation model, on both language, vision, and multi-modal tasks, and performs competitively with state-of-the-art models across a wide range of tasks and modalities. Moreover, DAVINCI also shows strong few-shot and zero-shot image/text generation capability.

In addition, most existing VLMs are pre-trained with mixed pre-training objectives and different data sources varying in size, making it difficult to disentangle the impact of pre-training objectives and data sources on the downstream tasks. To this end, we conduct a systematic analysis of the performance of generative vision-language pre-training by carefully ablating different pre-training objectives, such as prefix language / image modeling, and the amount of pre-training data with different qualities, revealing the impact of different objectives and data sources to facilitating future research.

To summarize, our contribution is three-fold: (1) We introduce prefix multi-modal modeling, a simple unified generative vision-language pre-training framework that is scalable for large-scale pre-training and versatile for image-to-text generation, text-to-image generation and various multi-modal / single-modal understanding tasks. (2) We pre-train DAVINCI, a vision-language foundation model, with the proposed approach, demonstrating competitive performance on a wide range of 27 downstream tasks and the superiority of combining vision/language generative pre-training. (3) We conduct an analysis about the impact of different pre-training data sources and pre-training objectives on the performance of seq2seq VLMs.

## 2 RELATED WORK

Inspired by the success of language model pre-training, several studies investigated vision-language pre-training on large-scale image-caption pairs. ViLBERT (Lu et al., 2019) and LXMERT (Tan & Bansal, 2019b) first propose to extract visual object features with an external object detection model like Fast-RCNN (Girshick, 2015), feed the image features together with texts into Transformer

---

[2]Named after the Italian polymath *Leonardo da Vinci*, who displayed infinite grace in everything. We noticed that this name is used in GPT-3 versioning. However, we think there is no conflict because it is only a suffix for a specific checkpoint of the GPT-3 family.

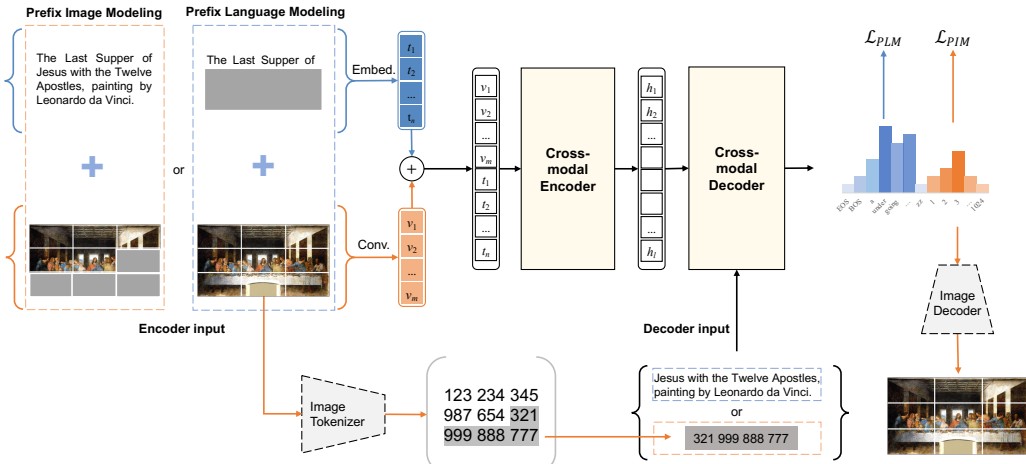

**Figure 1:** Illustration of the overall architecture and pre-training procedures of DAVINCI, a Transformer-based sequence-to-sequence model. Given an image-text pair, DAVINCI first splits either the word sequence or image token sequence into prefix and suffix. It then concatenates the prefix with the complete sequence in the other modality as input. DAVINCI is trained to recover the suffix with maximum likelihood estimation.

models, and train the model to align vision and language representations with masked multi-modal modeling and multi-modal alignment prediction objectives. Many following works (Li et al., 2020; Zhang et al., 2021b; Chen et al., 2020; Li et al., 2022a; 2021; Zeng et al., 2021; Wang et al., 2021a) propose several new objectives to improve object detection based VLP and explored using vision Transformer (Dosovitskiy et al., 2021; Touvron et al., 2021) as visual feature extractor.

More recently, FLAVA (Singh et al., 2021), a new vision-language foundation model, is pre-trained with a masked multi-modal modeling objective. Performing competitively on language, vision, and vision-language understanding tasks, FLAVA is designed for understanding tasks without text and image generation abilities.

While achieving promising results on **multi-modal understanding** tasks, most VLMs are based on encoder-only architectures with bidirectional attention, making them non-trivial to adapt to **multi-modal generation** tasks such as image captioning and text-to-image generation. Inspired by the success of seq2seq pre-trained language models such as T5 (Raffel et al., 2020) and BART (Lewis et al., 2020), VL-T5 (Cho et al., 2021) and OFA (Wang et al., 2022) propose to formulate both vision-language pre-training objectives and various downstream vision-language tasks as seq2seq tasks and pre-train a seq2seq VLM by multi-tasking on these tasks. However, the scalability and the zero-shot transfer capability of this approach are limited by the availability of large-scale and diverse vision-language tasks. To this end, SimVLM (Wang et al., 2021b), the most related work to our approach, instead pre-trains a seq2seq VLM with a simple prefix language modeling objective on text generation. It easily scales to very large and potentially noisy pre-training data and achieves competitive results. However, SimVLM only exploits language self-supervision, and thus it does not perform well on image understanding tasks and is unable to tackle image generation tasks. Another recent study is CM3 (Aghajanyan et al., 2022), which proposes a causal masked multi-modal model learned from large web data and differs from our work in pre-training objectives and target tasks.

As for the text-to-image generation task, Ramesh et al. (2021); Ding et al. (2021); Yu et al. (2022) achieved promising performance by learning an auto-regressive target with Transformer and VQ-VAE / VQ-GAN tokenizer. Most recently, Ramesh et al. (2022); Saharia et al. (2022) advanced the image generation capability by using diffusion models and high-quality text embeddings (e.g., CLIP, T5). Therefore, it is natural to explore boosting image generation via stronger multi-modal understanding.

Previous studies are good at either image-to-text or text-to-image generation, but few studies investigate whether these two important capabilities can be learned together and boost each other. In this paper, we explore making a versatile and powerful multi-modal foundation model that is good at text-to-image generation, image-to-text generation, and multi-modal understanding tasks.

## 3 DAVINCI

Given the superior performance of auto-regressive language models (LM) (Brown et al., 2020; Chowdhery et al., 2022; Rae et al., 2021) on zero-shot and few-shot transfer abilities, we decided to

adopt a decoder optimized by language modeling loss to retain the generalization capabilities, and an encoder to represent the prefix input. Unlike using a causal mask in the decoder, the encoder employs fully-visible attention for the prefix input. This architecture resembles prefix language modeling, which shows effectiveness in a wide range of language tasks (Dong et al., 2019; Raffel et al., 2020) and enables zero-shot generalization abilities. Contrary to the previous multi-stage approaches (Wang et al., 2021a; Singh et al., 2021), our model is trained from scratch in an end-to-end manner thanks to the model's simplicity. In this section, we introduce the proposed prefix multi-modal modeling framework and the DAVINCI model. The overall architecture of DAVINCI is depicted in Figure 1. We first explain our model architecture in detail in §3.1 and then introduce pre-training objectives and procedures in §3.2.

## 3.1 Model Architecture

**Textual Feature Embedding** Given an input sentence $S$, we first use WordPiece (Wu et al., 2016) to tokenize it to a sequence of tokens $W = \{w_1, w_2, ..., w_n\}$. To obtain text features $T$, for each token $w_i$, a token embedding $e_i$ and position embedding $p_i$ are computed by two separate embedding matrices. Finally, the textual feature embedding $T = \{t_1, t_2, ..., t_i, ..., t_n\}$ is calculated by $t_i = LayerNorm(e_i + p_i)$, where $i$ indicates the $i$-th position, and $LayerNorm$ (Ba et al., 2016) is a layer normalization function.

**Visual Feature Embedding** Given an input image $I$, we first use a CNN backbone to extract and learn the image features. Following (Dai et al., 2021; Wang et al., 2021b), we use the first three blocks of ResNet (He et al., 2016) to obtain the feature maps. The feature maps are then flattened to $F = \{f_1, f_2, ..., f_m\}$ along the spatial dimension, where $m$ denotes the number of features. To keep the position information of visual features, we inject absolute learned positional embeddings $p$ and the final visual embeddings $V = \{v_1, v_2, ..., v_i, ..., v_m\}$ are calculated by $v_i = f_i + p_i$, where $i$ indicates the $i$-th position.

**Cross-Modal Transformer** To fuse the textual and visual feature embeddings into a common space, we adopt a simple canonical Transformer architecture as the fusion module. The input is the combination of visual embedding $V$ and textual embedding $T$, namely $X = \{x_1, x_2, ..., x_l\} = [V, T] = \{v_1, v_2, ..., v_m, t_1, t_2, ..., t_n\}$. The input embedding vectors $X$ are then fed into a cross-modal Transformer encoder to obtain hidden state vectors $H = \{h_1, h_2, ..., h_l\}$. Finally, a Transformer decoder is applied to generate visual or textual tokens with $H$ and decoder input as illustrated in Figure 1.

**Image Tokenizer and Decoder** Because Transformer is modeling on discrete tokens, to unify the text tokens and image tokens, we discretize an image into tokens by an image tokenizer and reconstruct the raw image by an image decoder. The image tokenizer and decoder are implemented with a discrete variational autoencoder (dVAE) (Ramesh et al., 2021). After training of the image tokenizer, it could tokenize an image $I$ into a sequence of discrete visual tokens $Z = \{z_1, z_2, ..., z_m\}$ according to a learned vocabulary. Visual tokens $Z$ serve as the ground-truth labels for the prefix image modeling objective. In our work, we directly use an off-the-shelf image tokenizer and decoder from VQGAN (Esser et al., 2021), with a vocabulary size of 1024 and a compression rate of 16, which means a $256 \times 256$ image will be tokenized into $16 \times 16$ grid of tokens and then flattened to a sequence of 256 tokens.

## 3.2 Pre-training Objectives

Our major motivation is to conduct language modeling with image information and image modeling with text information simultaneously, which only requires image and text pairs that are easy to collect, making our approach easy to scale. The interaction would force the vision-language model to have a deeper understanding of both text and image. Learning from this interaction connects the visual representation with textual representation, enabling zero-shot transfer.

**Prefix Language Modeling (PLM)** The core idea of prefix language modeling is "given a full image $X_{image}$ and a prefix caption $\tilde{X}_{text}$, recover the masked textual tokens (i.e., suffix caption $Y_{text}$)". Given an input caption, we first randomly mask some continuous words at the end (we call it suffix caption hereafter) and recover the masked textual tokens with full image by optimizing the cross-entropy loss,

$$\mathcal{L}_{\text{PLM}} = - \sum_{(I,S) \in D} \log p(\mathbf{Y}_{\text{text}} | \mathbf{X}_{\text{image}}, \tilde{\mathbf{X}}_{\text{text}}), \qquad (1)$$

where I and S are images and captions from the pre-training corpus $D$.

Because of the lack of textual information, recovering the suffix caption requires the model to understand both the image and prefix caption. The full image is rich in semantic information that would help language modeling. The prefix length is randomly decided during training, and especially when prefix caption is none, this task will degenerate into "image captioning" task, which forces the model to generate a caption with the input image.

$$\mathcal{L}'_{\text{PLM}} = - \sum_{(I,S)\in D} \log p(\mathbf{Y}_{\text{text}} | \mathbf{X}_{\text{image}}) \qquad (2)$$

**Prefix Image Modeling (PIM)** The core idea of prefix image modeling is "given a full caption and a corrupted image (we call it prefix image hereafter), recover the masked visual tokens". Given an input image, we first randomly mask some continuous image patches at the end (we call it suffix image hereafter). The prefix image and full caption will be fed into the model and try to recover the original visual tokens obtained from the image tokenizer by optimizing the cross-entropy loss.

$$\mathcal{L}_{\text{PIM}} = - \sum_{(I,S)\in D} \log p(\mathbf{Y}_{\text{image}} | \mathbf{X}_{\text{text}}, \tilde{\mathbf{X}}_{\text{image}}) \qquad (3)$$

Similar to PLM, when prefix image is none, this task will degenerate into "text-to-image generation" task, forcing the model to generate an image with the input caption:

$$\mathcal{L}'_{\text{PIM}} = - \sum_{(I,S)\in D} \log p(\mathbf{Y}_{\text{image}} | \mathbf{X}_{\text{text}}) \qquad (4)$$

**Unified Learning Objective** Our model is learned by optimizing the combination of PLM and PIM.

$$\mathcal{L} = \mathcal{L}_{\text{PLM}} + \mathcal{L}_{\text{PIM}} \qquad (5)$$

## 4 EXPERIMENTS

### 4.1 PRE-TRAINING DATASETS

Since existing studies pre-trained their models on different corpora, making the fair comparison difficult. Considering results only on state-of-the-art performance would underestimate the potential of this line of research. Therefore, we propose several practical settings including small-scale and large-scale, and then conduct detailed comparisons on them in Section 5.1. More details about the datasets are shown in Appendix A.3.

| Data Type | Dataset | Image Domain | #Total |
|---|---|---|---|
| In-Domain Data (ID) | COCO, Visual Genome | COCO | 1.3M |
| Small-scale Web Data (SWD) | SBU, CC-3M, CC-12M | Web | 14.9M |
| Object-Region Data (ORD) | VG regions, VG objects, COCO objects, Refcoco, Open Image, Obj365 | COCO, Flickr | 17.0M |
| Vision Data (VD) | ImageNet-21K | ImageNet | 13.2M |
| Large-scale Web Data (LWD) | LAION-400M, DAVINCI-200M | Web | 601.3M |
| Text Data (TD) | C4 | Web | 800GB |

**Table 1: Statistics of the pre-training datasets.** #Total denotes the total number of image-text pairs.

### 4.2 DOWNSTREAM TASKS

We test our models' ability and versatility on five dimensions: **language understanding** on 8 GLUE tasks (Wang et al., 2019), **vision understanding** on ImageNet fine-tuning and 12 popular vision datasets for linear evaluation, **multi-modal understanding** on VQAv2 (Goyal et al., 2017b), SNLI-VE (Xie et al., 2019) and NLVR2 (Suhr et al., 2019), **text-to-image generation** on COCO (Chen et al., 2015), and **image-to-text generation** on COCO, NoCaps (Agrawal et al., 2019), and VLUE (Zhou et al., 2022b). Details of downstream tasks and fine-tuning process are described in Appendix A.2.

### 4.3 IMPLEMENTATION DETAILS

Our model is a base-size Transformer implemented with a 6-layer encoder and a 6-layer decoder, 768 dimensions for hidden states, 512 for maximum input length, and 3072 for intermediate size. We train our model from scratch without initializing the Transformer encoder and decoder. However, the image encoder is initialized from ResNet-101 (He et al., 2016) with ImageNet weights since we find

| Task | Eval. | BERT | RoBERTa | ViT | MLM 1 | MIM 2 | FLAVA 3 | CLIP 4 | SimVLM 5 | DAVINCI 6 | SimVLM 7 | DAVINCI 8 |
|---|---|---|---|---|---|---|---|---|---|---|---|---|
| | | 16GB | 160GB | 13.2M | 70M | 70M | 70M | 70M | 46.4M | 46.4M | 647.7M | 647.7M |
| MNLI | FT | 84.20 | 87.60 | – | 73.23 | – | 80.33 | 32.85 | 82.13 | 82.25 | **83.27** | 83.13 |
| CoLA | FT | 54.60 | 63.60 | – | 39.55 | – | 50.65 | 11.02 | 52.47 | 52.10 | 54.22 | **54.75** |
| MRPC | FT | 84.75 | 90.20 | – | 73.24 | – | 84.16 | 68.74 | 82.70 | 83.14 | 84.26 | **84.54** |
| QQP | FT | 89.00 | 91.90 | – | 86.68 | – | 88.74 | 59.17 | 88.39 | 88.15 | **89.05** | 88.92 |
| SST-2 | FT | 92.50 | 94.80 | – | 87.96 | – | 90.94 | 83.49 | 90.65 | 90.48 | 91.12 | **91.37** |
| QNLI | FT | 91.00 | 92.80 | – | 82.32 | – | 87.31 | 49.46 | 87.55 | 87.21 | **88.28** | 87.90 |
| RTE | FT | 62.50 | 78.70 | – | 50.54 | – | 57.76 | 53.07 | 59.80 | 60.72 | 63.34 | **64.22** |
| STS-B | FT | 88.20 | 91.20 | – | 78.89 | – | 85.67 | 13.70 | 86.62 | 86.27 | **87.24** | 87.05 |
| **NLP Avg.** | | 80.84 | 86.35 | – | 71.55 | – | 78.19 | 46.44 | 78.79 | 78.79 | 80.10 | **80.23** |
| ImageNet | LE | – | – | 80.90 | – | 41.79 | 75.54 | 72.95 | 74.31 | 75.87 | 76.04 | **77.65** |
| Food101 | LE | – | – | 86.70 | – | 53.30 | 88.51 | 85.49 | 83.41 | 89.33 | 85.52 | **90.12** |
| CIFAR10 | LE | – | – | 96.90 | – | 76.20 | 92.87 | 91.25 | 91.56 | 93.01 | 92.41 | **93.96** |
| CIFAR100 | LE | – | – | 86.40 | – | 55.57 | 77.68 | 74.40 | 72.51 | 78.98 | 75.23 | **80.11** |
| Cars | LE | – | – | 54.70 | – | 14.71 | 70.87 | 62.84 | 61.44 | 72.69 | 68.83 | **74.57** |
| Aircraft | LE | – | – | 46.00 | – | 13.83 | 47.31 | 40.02 | 41.28 | 47.42 | 47.75 | **49.55** |
| DTD | LE | – | – | 74.30 | – | 55.53 | 77.29 | 73.40 | 72.55 | 77.12 | 76.59 | **78.33** |
| Pets | LE | – | – | 92.70 | – | 34.48 | 84.82 | 79.61 | 78.77 | 85.52 | 86.13 | **88.21** |
| Flowers102 | LE | – | – | 99.20 | – | 67.23 | 96.37 | 94.94 | 93.24 | 96.12 | 95.41 | **96.88** |
| MNIST | LE | – | – | 97.40 | – | 96.40 | 98.42 | 97.38 | 96.66 | 98.67 | 98.45 | **99.01** |
| STL10 | LE | – | – | 99.50 | – | 80.12 | 98.89 | 97.29 | 97.51 | 99.03 | 98.02 | **99.21** |
| Country211 | LE | – | – | 17.50 | – | 8.87 | 28.92 | 25.12 | 26.45 | 28.99 | 27.81 | **29.94** |
| **Vision Avg.** | | – | – | 77.68 | – | 49.84 | 78.12 | 74.56 | 74.14 | 78.56 | 77.34 | **79.80** |
| VQAv2 | FT | – | – | – | – | – | 72.49 | 59.81 | 72.12 | 73.89 | 75.03 | **76.44** |
| SNLI-VE | FT | – | – | – | – | – | 78.89 | 73.53 | 78.74 | 79.11 | 79.63 | **80.01** |
| NLVR2 | FT | – | – | – | – | – | – | – | 77.45 | 77.91 | 79.72 | **80.25** |
| I2T@B4 | FT | – | – | – | – | – | – | – | 38.00 | 38.50 | 38.10 | **39.20** |
| I2T@C | FT | – | – | – | – | – | – | – | 126.96 | 128.66 | 128.91 | **130.44** |
| T2I@IS ↑ | FT | – | – | – | – | – | – | – | – | 17.55 | – | **22.41** |
| T2I@FID ↓ | FT | – | – | – | – | – | – | – | – | 23.58 | – | **19.82** |
| VQAv2 | FS | – | – | – | – | – | – | – | 54.69 | 54.85 | 51.88 | **54.90** |
| SNLI-VE | FS | – | – | – | – | – | – | – | 67.45 | 67.57 | 67.96 | **68.04** |
| NLVR2 | FS | – | – | – | – | – | – | – | 51.46 | 51.19 | 51.49 | **51.52** |
| I2T@B4 | FS | – | – | – | – | – | – | – | 35.90 | 36.40 | 32.70 | **37.00** |
| I2T@C | FS | – | – | – | – | – | – | – | 117.75 | 120.43 | 112.20 | **122.56** |
| I2T@B4 | ZS | – | – | – | – | – | – | – | 11.40 | 10.80 | 13.80 | **18.70** |
| I2T@C | ZS | – | – | – | – | – | – | – | 45.30 | 45.55 | 56.69 | **68.44** |
| VLUE@B4 | ZS | – | – | – | – | – | – | – | 9.20 | 9.40 | 10.40 | **10.60** |
| VLUE@C | ZS | – | – | – | – | – | – | – | 33.92 | 34.80 | 39.75 | **40.83** |
| NoCaps@C | ZS | – | – | – | – | – | – | – | 48.05 | 45.51 | 48.64 | **58.58** |
| T2I@IS ↑ | ZS | – | – | – | – | – | – | – | – | 14.91 | – | **17.44** |
| T2I@FID ↓ | ZS | – | – | – | – | – | – | – | – | 29.83 | – | **24.21** |
| **Multi-modal Avg.** | | – | – | – | – | – | – | – | 57.89 | 58.30 | 59.13 | **62.50** |

Table 2: **Experimental results on vision, language and multi-modal downstream tasks.** @B4, @C denote BLEU@4, CIDEr, respectively. I2T and T2I denote image-to-text and text-to-image tasks. Multi-modal Avg. is the average score of all multi-modal tasks. FT: fine-tuning, LE: linear evaluation, FS: few-shot, ZS: zero-shot. Under few-shot setting, we fine-tune a pre-trained model for 3 epochs on 1% training data. Results for BERT are obtained from Iki & Aizawa (2021). Results for RoBERTa are from its corresponding paper (Liu et al., 2019) and they use the mid-training (Phang et al., 2018) on MNLI for RTE, MRPC and STS-B while other models (e.g., BERT, SimVLM, DAVINCI) do not apply this trick. Results for ViT are from ViT-Base/16 model (Radford et al., 2021). We list the reported performance of text-only and image-only models in grey for reference.

a warm start provides a reliable visual representation and helps the convergence. All pre-training experiments are conducted on 32GB NVIDIA V100 GPUs. The model trained on the largest data takes around 10 days on 1024 V100 GPUs. We adopt dynamic masking in our experiments, where the masking ratio is randomly sampled from a uniform distribution U(0, 1). More details of the fine-tuning, network architectures, and hyper-parameters setups are given in Appendix A.1.

## 4.4 EXPERIMENTAL RESULTS

We extensively compare the performance of DAVINCI with state-of-the-art unified foundation models and vision-language models across vision, language, and multi-modal tasks, accessing five different abilities: (1) text understanding, (2) image understanding, (3) text-to-image generation, (4) image-to-text generation, (5) multi-modal understanding.

**Overall Performance** We report the overall performance on 8 language tasks from GLUE, 12 vision tasks, 3 multi-modal tasks, 3 image-to-text tasks and 1 text-to-image task. We compare our model with FLAVA and SimVLM [3], two of the most recent and best performing vision-language

---

[3]Since SimVLM is not open-sourced and uses 1.8B in-house data without telling the exact size of its $base$ model, we replicate it on our data with the same size as DAVINCI. Experiments on SimVLM$_{small}$ ensure our successful reproduction (see Appendix A.4).

| Model | #Params. | Text MNLI Acc | Vision ImageNet LE / FT | Image2Text COCO B@4 / C | Text2Image COCO IS↑ / FID↓ | Multi-modal VQA test-dev / test-std | NLVR2 dev / test-P |
|---|---|---|---|---|---|---|---|
| *Encoder-only Multi-modal Models* | | | | | | | |
| VinVL (Zhang et al., 2021b) | 157M | – | – | 38.2 / 129.3 | – | 75.95 / 76.12 | 82.05 / 83.08 |
| ViLT (Kim et al., 2021) | 88M | – | – | – | – | 70.85 / – | 74.91 / 75.57 |
| ALBEF (Li et al., 2021) | 210M | – | – | – | – | 75.84 / 76.04 | 82.55 / 83.14 |
| X-VLM (Zeng et al., 2021) | 240M | – | – | 39.6 / 132.6 | – | 78.22 / 78.37 | 84.41 / 84.76 |
| VLMO (Wang et al., 2021a) | – | – | – | – | – | 76.64 / 76.89 | 82.77 / 83.34 |
| *Encoder-Decoder Multi-modal Models* | | | | | | | |
| UNICORN (Yang et al., 2021) | – | – | – | 35.8 / 119.1 | – | 69.20 / 69.40 | – / – |
| Uni-ENDN (Li et al., 2022b) | 110M | – | – | – | – | 72.20 / 72.50 | – / – |
| Pixel-BERT (Huang et al., 2020) | 144M | – | – | – | – | 74.45 / 74.55 | 76.50 / 77.20 |
| E2E-VLP (Xu et al., 2021a) | 94M | – | – | 36.2 / 117.3 | – | 73.25 / 73.67 | 77.25 / 77.96 |
| VL-T5 (Cho et al., 2021) | 220M | – | – | 34.5 / 116.5 | – | – / 70.30 | 74.60 / 73.60 |
| VL-BART (Cho et al., 2021) | 220M | – | – | 35.1 / 116.6 | – | – / 71.30 | 71.70 / 70.30 |
| *Text2Image Models* | | | | | | | |
| DM-GAN (Zhu et al., 2019) | – | – | – | – | 32.20 / 26.50 | – / – | – / – |
| DALLE (Ramesh et al., 2021) (250M) | 12B | – | – | – | 17.90 / 27.50 | – / – | – / – |
| DALLE (Ramesh et al., 2021) (640M)† | 82M | – | – | – | 15.79 / 29.22 | – / – | – / – |
| CogView (Ding et al., 2021) | 4B | – | – | – | 18.20 / 27.10 | – / – | – / – |
| *Unified Models* | | | | | | | |
| Unifying (Huang et al., 2021) | 228M | – | – | 37.3 / 122.6 | – / 29.90 | – / – | – / – |
| FLAVA (Singh et al., 2021) | 240M | 80.33 | 75.54 / – | – | – | 72.80 / 72.49 | – / – |
| SimVLM (Wang et al., 2021b) (640M)† | 153M | **83.27** | 76.04 / – | 38.5 / 128.7 | – | 75.04 / 75.03 | 78.82 / 79.72 |
| SimVLM (Wang et al., 2021b) (1.8B) | | 83.40 | 80.60 / – | 39.0 / 134.8 | – | 77.87 / 78.14 | 81.72 / 81.77 |
| OFA (Wang et al., 2022) | 182M | 84.30 | – / 82.20 | 41.0 / 138.2 | 21.50* / 20.80* | 78.00 / 78.10 | – / – |
| Florence (Yuan et al., 2021) | 637M | – | – / 90.05 | – / – | – / – | 80.16 / 80.36 | – / – |
| DAVINCI | 154M | 83.13 | **78.81 / 83.92** | **39.2 / 130.4** | **17.44 (22.41*) / 24.21 (19.82*)** | 76.32 / 76.44 | 80.03 / 80.25 |

**Table 3: Comparison with state-of-the-art vision-language models on vision, language, and multi-modal downstream tasks.** All results are from *base*-size models. LE and FT denote linear evaluation and fine-tuning performance, respectively. Image2Text results are reported without CIDEr optimization. † are our reproduced models. * are the results after fine-tuning. SimVLM (1.8B) and OFA are pre-trained with much larger corpus or human-labeled data of many downstream tasks, and thus they are not comparable and are labeled in gray. Florence (Yuan et al., 2021) is pre-trained with much larger model size (Florence-CoSwin-H, 637M) and more pre-training data (900M), so the numbers are in grey. **bold** denotes the best across unified models.

foundation models. We also include comparisons with some baseline models (e.g., MIM, MLM, CLIP). There are several observations. First, DAVINCI (column 8) outperforms FLAVA (column 3) and SimVLM (column 7) across almost all tasks, providing a new and stronger unified foundation model. Compared with FLAVA, DAVINCI improves an average of 2.04%, 1.68% on language and vision tasks, respectively. Compared with SimVLM, DAVINCI achieves comparable results on language tasks (+0.13%) while performing much better on vision tasks (+2.46%). To make a fair comparison in terms of similar data size, we compare FLAVA (70M data, column 3) with DAVINCI (46.4M data, column 6). It is observed that DAVINCI still outperforms FLAVA even with much less data. Considering the multi-modal tasks, DAVINCI consistently outperforms FLAVA and SimVLM on VQA and VE. Note that FLAVA is incapable of generation and SimVLM cannot generate images; only DAVINCI is competent to all tasks and demonstrates a stronger capability of unifying vision and language tasks.

**Zero-shot and Few-shot Transfer** One of the critical benefits of generative pre-trained vision-language models is the good generalization ability on zero-shot and few-shot tasks. For zero-shot transfer, two out-of-domain distribution datasets are considered (NoCaps and VLUE), with results shown in Table 2. First, DAVINCI outperforms SimVLM on both zero-shot and few-shot settings, demonstrating its better transfer capabilities. It also shows the effectiveness and robustness of the synergy of our proposed language supervision and image supervision. Second, it is observed that the performance improvement is bigger on 647.7M data (column 7 v.s. column 8) than 46.4M data (column 5 v.s. column 6). This shows DAVINCI generalizes well with the increase of large-scale data. We even observe some performance drops on small data (46.4M) but excellent performance improvements on large data (647.7M). It is consistent with the recent observation that zero-shot ability could only be triggered with large pre-training data (Wei et al., 2022) and scaling to large data and keeping simple training objectives benefit generalization performance (Wang et al., 2021b).

**Comparison with state-of-the-art vision-language models** In addition to unified vision-language foundation models, we compare DAVINCI with state-of-the-art vision-language models as well. The results are shown in Table 2. DAVINCI demonstrates its superiority in vision understanding and text-to-image generation. Compared with current popular auto-regressive image generation models like DALLE and CogView, our model achieves comparable IS and better FID scores with significantly fewer model parameters than DALLE and CogView. Note that the original DALLE is implemented based on VQVAE, so here, we compare our model with reproduced VQGAN-based DALLE with

| Settings | Pre-training Data | | | | | #Image | #Caption | Models | COCO Captions | VQA | SNLI-VE | NLVR2 |
|---|---|---|---|---|---|---|---|---|---|---|---|---|
| | ID | SWD | ORD | VD | LWD | | | | B@4 / C | Acc | Acc | Acc |
| 1 | ✓ | | | | | 0.2M | 1.3M | SimVLM | 35.2 / 115.06 | 68.89 | 76.10 | 71.21 |
| | | | | | | | | DaVinci | 35.8 / 117.30 | 69.25 | 76.22 | 72.55 |
| 2 | ✓ | ✓ | | | | 15.1M | 16.2M | SimVLM | 37.0 / 122.63 | 71.54 | 78.36 | 75.50 |
| | | | | | | | | DaVinci | 37.4 / 123.11 | 71.88 | 78.62 | 77.46 |
| 3 | ✓ | | ✓ | | | 2.7M | 18.3M | SimVLM | 38.2 / 123.85 | 69.57 | 76.65 | 70.50 |
| | | | | | | | | DaVinci | 38.0 / 124.20 | 70.02 | 76.92 | 72.01 |
| 4 | ✓ | | | ✓ | | 13.4M | 14.5M | SimVLM | 36.2 / 119.73 | 70.53 | 76.90 | 73.25 |
| | | | | | | | | DaVinci | 36.6 / 121.27 | 71.23 | 77.40 | 74.62 |
| 5 | ✓ | ✓ | ✓ | ✓ | | 30.5M | 46.4M | SimVLM | 38.5 / 128.12 | 71.84 | 78.81 | 76.75 |
| | | | | | | | | DaVinci | 38.6 / 128.73 | 73.53 | 79.24 | 77.55 |
| 6 | | | | | ✓ | 601.3M | 601.3M | SimVLM | 37.3 / 123.81 | 73.73 | 78.79 | 77.69 |
| | | | | | | | | DaVinci | 37.6 / 124.42 | 73.95 | 79.29 | 78.54 |
| 7 | ✓ | | | | ✓ | 601.5M | 602.6M | SimVLM | 37.9 / 125.50 | 74.64 | 79.05 | 77.68 |
| | | | | | | | | DaVinci | 38.1 / 125.91 | 74.91 | 79.22 | 78.12 |
| 8 | ✓ | ✓ | ✓ | ✓ | ✓ | 631.8M | 647.7M | SimVLM | 38.5 / 128.25 | 75.04 | 79.32 | 78.82 |
| | | | | | | | | DaVinci | 39.1 / 130.21 | 76.32 | 80.04 | 80.03 |

Table 4: **Evaluation on downstream tasks using COCO Captions, VQA, SNLI-VE, and NLVR2.** #Image and #Caption denote the numbers of images and image-text pairs that are used in the pre-training.

similar model sizes, and find DaVinci still achieves a significant improvement over it. Generated images are presented in Appendix A.11 for further qualitative comparison.

On multi-modal tasks such as VQA, DaVinci not only outperforms unified models (e.g., SimVLM (640M)) and other encoder-decoder multi-modal models (e.g., E2E-VLP, VL-T5), but also achieves competitive performance with many conventional encoder-only multi-model models (e.g., VinVL, ALBEF, VLMO). Note that SimVLM (1.8B) and OFA are not directly comparable because SimVLM uses 1.8B in-house image-text pairs, and OFA uses human-labeled data of many downstream tasks during pre-training. Even though, we still report their results for reference and observe a better performance on ImageNet fine-tuning and text-to-image generation than OFA.

The advantages of image generation over DALLE / CogView, the superiority of image-to-text over SimVLM, and the competitive performance with conventional multi-modal models demonstrate the synergistic effect of our proposed PLM (language supervision) and PIM (image supervision).

## 5 ANALYSIS

### 5.1 IMPACT OF PRE-TRAINING DATASETS

In this section, we disclose the impact of various multi-modal data sources for VLMs. We choose SimVLM and DaVinci as our baseline models for their competitive performance, the capability of training from scratch, and the scalability of extending to the noisy large-scale corpus. We use the same text corpus, $C4$, for all the variations. The results are shown in Table 4. In general, the performance is increased along with the data size, and DaVinci consistently outperforms SimVLM on almost all the data settings and all the downstream tasks. Both object-region data and vision data are clearly helpful in vision language pre-training (refer to settings 3 and 4). We surprisingly observe that models pre-trained on object-region data with much fewer images performs even better than models pre-trained with small-scale web data on the COCO Caption task (refer to settings 2 and 3). Although large-scale web data is usually noisier than small datasets (e.g., ID, ORD, VD, and SWD), it is powerful for multi-modal pre-training (refer to settings 5 and 8). We believe our analysis has broader impacts on the research of VLMs in the community. First, this enables fair comparisons for pre-trained models in the same data settings. Second, one can focus on the model designs at part or all of the data settings according to available computation resources. Third, we reveal that object-region and vision data, normally overlooked in VLM pre-training, also play a significant role.

### 5.2 ABLATION STUDY

To verify the contributions of different modules in our framework, we ablate them and evaluate DaVinci on five kinds of downstream tasks: language understanding (MNLI, SST-2), vision understanding (ImageNet, Food101, CIFAR10), multi-modal understanding (VQAv2, SNLI-VE, NLVR2), image-to-text generation (COCO Captions), and text-to-image generation. Experiments are conducted with the same model architecture on in-domain data (ID). The results are shown in Table 5.

**Effects of Objectives** First, all three objectives (PLM, PIM, and Text2Text) bring improvement and the combination confirms a synergistic effect. Second, it is observed that without PLM, the performance decreases significantly on multi-modal understanding and image-to-text generation,

| Method | COCO B@4 / C | VQA Acc | SNLI-VE Acc | NLVR2 Acc | ImageNet Acc | Food101 Acc | CIFAR10 Acc | MNLI Acc | SST-2 Acc | T2I IS / FID |
|---|---|---|---|---|---|---|---|---|---|---|
| No Pre-training | 32.1 / 96.71 | 52.73 | 54.23 | 51.08 | –* | –* | –* | 66.32 | 79.84 | –* |
| DaVinci | **35.8 / 117.30** | **69.25** | **76.22** | **72.55** | **48.88** | **75.32** | **73.82** | 81.76 | 90.25 | **12.35 / 53.14** |
| – PLM | 33.6 / 111.17 | 65.15 | 73.91 | 53.28 | 48.05 | 74.17 | 72.98 | 81.42 | 89.97 | 10.26 / 59.64 |
| – PIM | 34.3 / 116.58 | 68.89 | 75.79 | 69.78 | 45.54 | 71.18 | 70.11 | 81.94 | **90.53** | –* |
| – Text2Text | 34.1 / 115.21 | 68.14 | 75.38 | 70.34 | 48.67 | 74.26 | 73.23 | 76.48 | 88.14 | 12.07 / 54.77 |
| PL=0 | 35.4 / 117.00 | 66.90 | 75.52 | 71.05 | 48.45 | 68.18 | 73.73 | 78.69 | 89.00 | 11.76 / 55.38 |
| PL=15% | 35.7 / 116.53 | 69.16 | 75.09 | 70.44 | 41.58 | 52.15 | 68.55 | 79.02 | 89.46 | –* |
| PL=50% | 35.1 / 115.53 | 68.55 | 74.54 | 56.92 | 37.69 | 49.16 | 70.15 | 78.59 | 89.69 | –* |
| MIM | 34.7 / 113.4 | 68.18 | 75.34 | 69.66 | 48.46 | 56.95 | 72.79 | 81.72 | 89.84 | 9.50 / 74.13 |
| In-painting | 34.5 / 112.5 | 67.46 | 75.41 | 68.66 | 47.50 | 54.38 | 71.20 | 81.55 | 89.84 | 9.97 / 68.15 |
| Token Projection | 17.7 / 49.2 | 52.13 | 71.11 | 52.01 | 15.11 | 25.62 | 61.01 | **82.01** | 90.25 | 11.89 / 60.96 |
| Patch Projection | 25.7 / 79.5 | 57.69 | 71.92 | 57.45 | 36.23 | 44.31 | 69.40 | 81.73 | 90.05 | 11.41 / 61.87 |

**Table 5: Ablation study on COCO Captions, VQA, SNLI-VE, NLVR2, ImageNet, Food101, CIFAR10, MNLI, SST-2, and text-to-image (T2I) generation.** "–" denotes removing the corresponding objective. PL denotes the prefix length under fixed masking ratio settings. Because the linear probe requires a pre-trained model to be frozen, "No Pre-training" results on ImageNet, Food101, and CIFAR10 are not reported and labeled by *. For T2I, we report the zero-shot results. Note that the following four variants cannot perform zero-shot text-to-image generation (labeled by *): (1) No Pre-training, (2) DaVinci – PIM, (3) PL=15%, and (4) PL=50%.

indicating the importance of language supervision. Third, PIM brings more gains than PLM and text2text on vision understanding, which is expected because it enhances the vision encoding ability with image supervision. In addition, the text2text objective is important to text understanding. Last, on the text-to-image generation task, it is observed that PLM is also helpful, confirming the synergistic effect of PIM and PLM again. Intuitively, PIM and PLM can help each other learn the alignments of visual and textual features, which will benefit both image generation and other multi-modal tasks.

**Effects of Masking Ratios** Our model adopts dynamic masking ratios as described in Section 3.2. We also conduct experiments with static masking ratios with the prefix length fixed to 0, 15%, and 50%. The comparison between dynamic masking ratios and static masking ratios (PL=0, 15%, and 50%) reveals that dynamic masking is better. We attribute this improvement to the smoothing effects of dynamic masking ratios. We also find that the standard language model (PL=0) performs worse on VQA, Food101, and text-to-image generation, which is consistent with the observation in SimVLM. In our experiments, the masking ratio is sampled from a uniform distribution U(0, 1).

**Effects of Masking Strategies** Here we also compared three different masking strategies: 1) masked image modeling (randomly masking some patches), 2) in-painting (randomly masking some continuous spans in the middle of the image), and 3) suffix-painting (ours). The results are shown in Table 5. Both masked image modeling and in-painting are effective and competitive. It is observed that suffix-painting is better than masked image modeling and in-painting across all tasks, demonstrating that suffix-painting works well.

**Effects of Image Feature Extraction** There are several different ways to extract image features. We compare three different image representation methods: 1) token projection (projecting the prefix tokens to the hidden dimension of the backbone network on the token-level), 2) patch projection (similar to ViT embedding, we split an image into fixed-size patches, embed each of them by a trainable linear projection on the pixel-level), and 3) ResNet feature extraction (ours). From the results in Table 5, we observed that ResNet feature extraction outperforms token projection and patch projection by a large margin. Therefore, we decided to adopt ResNet to extract image features.

We provide more details and discussions about the effects of compute (A.5), masking strategies (A.6), image feature extraction methods (A.7), and scaling effects of data size ( A.8) in the Appendix.

# 6 CONCLUSION AND DISCUSSION

In this work, we first benchmark several settings on sequence-to-sequence vision-language pre-training in terms of pre-training dataset size, aligning SimVLM and our model on them. We propose a simple and unified generative pre-training model, DaVinci, to simultaneously leverage the language supervision and image supervision through two objectives under a unified framework: prefix language modeling and prefix image modeling. DaVinci is simple yet effective, demonstrating strong capabilities in both multi-modal writing and painting tasks. Experimental results explicitly imply that combining suffix caption generation and suffix image generation offers large gains on all benchmark settings. We also discussed limitations and future work in Appendix A.10.

## ACKNOWLEDGMENTS

We thank the anonymous reviewers for their valuable suggestions. We would like to acknowledge Yan Zeng, Wenguan Huang, and Zhi Zhang at ByteDance, and Zhiling Zhang at Shanghai Jiao Tong University for their generous assistance in data collection and helpful discussions. We also wish to thank Hang Li at ByteDance, and Tong Zhang at HKUST for inspiring feedback, valuable comments, and great support to this work.

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

# A APPENDIX

## A.1 DETAILS OF HYPER-PARAMETERS

**Pre-training** Our model is a base-size Transformer implemented with a 6-layer encoder and a 6-layer decoder, 768 dimensions for hidden states, 512 for maximum input length, and 3072 for intermediate size. We train our model from scratch without initializing the Transformer encoder and decoder. The image encoder is initialized from ResNet-101 (He et al., 2016) with ImageNet weights since we find a warm start provides a reliable visual representation and helps the convergence. For models pre-training on large-scale data, we optimize 10 epochs while for other small-scale datasets, we optimize 40 epochs with the AdamW optimizer. The weight decay is set to 0.01 with $\beta_1 = 0.9, \beta_2 = 0.999$. The learning rate is 2e-4 with a warm-up period for the first 2% steps and linearly decayed to 0 after 2% of the total training steps. In each batch, there are 8,192 image-text pairs for text-to-image generation and image-to-text generation with 8,192 text-only documents for text-to-text generation. We use center-crop to resize each image to the size of 256×256, which is the only data augmentation used during training. All pre-training experiments are conducted on 32GB NVIDIA V100 GPUs. We adopt mixed-precision (Micikevicius et al., 2018) to accelerate training and save memory. The model trained on the largest data takes around 10 days on 1024 V100 GPUs. The default settings are shown in Table 6. We adopt dynamic masking in our experiments, where the masking ratio is randomly sampled from a uniform distribution U(0, 1).

**Fine-tuning** The learning rate is $\in$ [1e-5, 5e-5] and our model is optimized by AdamW. Because the image resolution differs between pre-training and fine-tuning, the position parameters are adapted using linear interpolation. For all downstream tasks, we apply random resize crops and horizontal flips augmentation during training. All fine-tuning experiments are conducted on 32GB NVIDIA V100 GPUs. The default settings for text classification, image classification, multi-modal understanding and image-to-text generation are shown in Tables 7, 8, and 9, respectively.

| config | value |
|---|---|
| optimizer | AdamW (Loshchilov & Hutter, 2019) |
| learning rate | 2e-4 |
| weight decay | 0.01 |
| optimizer momentum | $\beta_1, \beta_2 = 0.9, 0.999$ |
| batch size | 8192 |
| learning rate schedule | linear decay |
| warmup ratio (Goyal et al., 2017a) | 0.02 |
| training epochs | {10, 40} |
| augmentation | RandomResizedCrop |

**Table 6:** Pre-training setting.

| config | value |
|---|---|
| optimizer | AdamW |
| learning rate | {1e-5, 2e-5, 5e-5} |
| weight decay | 0.01 |
| optimizer momentum | $\beta_1, \beta_2 = 0.9, 0.999$ |
| batch size | {16, 32, 64} |
| learning rate schedule | linear decay |
| warmup ratio | 0.1 |
| training epochs | {5, 10} |

**Table 7:** Text classification: GLUE setting.

## A.2 DETAILS OF DOWNSTREAM TASKS

**Language Understanding** We conduct experiments on GLUE benchmark including MNLI (Williams et al., 2018), CoLA (Warstadt et al., 2019), MRPC (Dolan & Brockett, 2005), QQP (Iyer et al., 2017), SST-2 (Socher et al., 2013), QNLI (Rajpurkar et al., 2016),

| config | value |
|---|---|
| optimizer | LARS (You et al., 2017) |
| base learning rate | 0.1 |
| weight decay | 0 |
| optimizer momentum | 0.9 |
| batch size | 16384 |
| learning rate schedule | cosine decay |
| warmup epochs | 10 |
| training epochs | 90 |
| augmentation | RandomResizedCrop |

**Table 8:** Image classification: Linear probing setting.

| config | value |
|---|---|
| optimizer | AdamW |
| learning rate | [1e-5, 5e-5] |
| weight decay | 0.02 |
| optimizer momentum | $\beta_1, \beta_2 = 0.9, 0.999$ |
| batch size | 1024 |
| learning rate schedule | linear decay |
| warmup epochs | [2, 5] |
| training epochs | [5, 15] |
| label smoothing (Szegedy et al., 2016) | 0.1 |
| augmentation | RandomResizedCrop, HorizontalFlips |

**Table 9:** Multi-modal understanding and image-to-text generation: fine-tuning setting.

RTE (Dagan et al., 2005; Haim et al., 2006; Giampiccolo et al., 2007; Bentivogli et al., 2009), and STS-B (Agirre et al., 2007). We follow the practice of BART (Lewis et al., 2020) and feed the same input to the encoder and decoder, and the hidden state of the final decoder token is fed into a new multi-class linear classifier or regression head. MNLI results are an average of MNLI-m and MNLI-mm. MRPC and QQP results are average of accuracy and F1. Matthews correlation coefficient (MCC) is reported for CoLA and Pearson correlation coefficient (PCC) is reported for STS-B.

**Vision Understanding** We conduct vision experiments in both fine-tuning and linear evaluation (linear eval). The linear evaluation follows a common practice (Caron et al., 2021; He et al., 2020; Singh et al., 2021) in self-supervised learning to evaluate the representation quality, where the pre-trained backbone model is frozen and a new linear classifier is appended on top of it. We choose 12 popular datasets: ImageNet (Russakovsky et al., 2015), Food101 (Bossard et al., 2014), CIFAR10 (Krizhevsky et al., 2009), CIFAR100 (Krizhevsky et al., 2009), Cars (Krause et al., 2013), Aircraft (Maji et al., 2013), DTD (Cimpoi et al., 2014), Pets (Parkhi et al., 2012), Flowers102 (Nilsback & Zisserman, 2008), MNIST (LeCun & Cortes, 2010), STL10 (Coates et al., 2011), and Country211 (Radford et al., 2021).

**Multi-modal Understanding** We consider three popular multi-modal tasks: VQAv2 (Goyal et al., 2017b), SNLI-VE (Xie et al., 2019) and NLVR2 (Suhr et al., 2019) to evaluate our model's multi-modal understanding ability. For VQAv2, following ALBEF (Li et al., 2021), the image and question are fed to the encoder and the decoder generates answers based on the multi-modal embeddings. For SNLI-VE, we follow SimVLM (Wang et al., 2021b) to feed the image to the encoder and the text to the decoder. A classifier is appended on top of our pre-trained model, and it is trained to predict the result based on the last hidden states of the decoder. For NLVR2, two input pairs are constructed, each of them including one image and the textual description. The prediction is made based on the concatenation of these two embeddings following SimVLM (Wang et al., 2021b). The resolutions for VQAv2, SNLI-VE, NLVR2 are 480, 384, 384, respectively.

**Text-to-Image Generation** The text-to-image task requires the model to understand the textual instruction first and then draw the image according to the input's intention. The input text is fed to our encoder, and our decoder will generate visual tokens one by one. After obtaining visual tokens, they are decoded into a raw image by an image decoder. We directly use an off-the-shelf image decoder from VQGAN (Esser et al., 2021). Following (Ramesh et al., 2021) we directly evaluate our

| Data Type | Dataset | Image Domain | #Images | #Captions | #Total |
|---|---|---|---|---|---|
| In-Domain Data (ID) | COCO | COCO | 110.3K | 551.7K | 1.3M |
| | Visual Genome | COCO | 108.2K | 759.0K | |
| Small-scale Web Data (SWD) | SBU | Web | 859.7K | 859.7K | 14.9M |
| | CC-3M | Web | 2.9M | 2.9M | |
| | CC-12M | Web | 11.1M | 11.1M | |
| Object-Region Data (ORD) | VG regions | COCO | 108.2K | 3.6M | 17.0M |
| | VG objects | COCO | 108.2K | 925.6K | |
| | COCO objects | COCO | 110.3K | 736.6K | |
| | Refcoco | COCO | 27.9K | 589.9K | |
| | Open Image | Flickr | 1.7M | 7.5M | |
| | Obj365 | Flickr | 577.6K | 3.6M | |
| Vision Data (VD) | ImageNet-21K | ImageNet | 13.2M | 13.2M | 13.2M |
| Large-scale Web Data (LWD) | DAVINCI-200M | Web | 205.6M | 205.6M | 601.3M |
| | LAION-400M | Web | 395.7M | 395.7M | |
| Text Data (TD) | C4 | Web | – | – | 800GB |

**Table 10: Statistics of the pre-training datasets.** #Images, #Captions, and #Total denote the number of images, the number of image-text pairs, and the total number of image-text pairs, respectively.

pre-trained model on 30, 000 images randomly sampled from COCO (Chen et al., 2015) validation split. Both Fréchet Inception Distance (FID) (Heusel et al., 2017) and Inception Score (IS) (Salimans et al., 2016) are reported. The image resolution is 256.

**Image-to-Text Generation** For image-to-text generation (also called image captioning), the image is given to encoder and the decoder will generate the corresponding caption. Our experiments are conducted on COCO dataset (Chen et al., 2015) with cross-entropy optimization. Other task-specific techniques such as CIDEr optimization (Rennie et al., 2017) are not introduced. The image resolution is 480. We also conduct zero-shot captioning experiments on NoCaps (Agrawal et al., 2019) and VLUE (Zhou et al., 2022b).

## A.3 PRE-TRAINING DATASETS

Since existing studies pre-trained their models on different corpora, some of which are publicly available (e.g., CC-3M, CC-12M) while some are in-house datasets (e.g., ALIGN (Jia et al., 2021)), making the fair comparison difficult. Considering results only on the state-of-the-art performance would underestimate the potential of this line of research. Therefore, we propose several practical settings, including small-scale and large-scale, and then conduct detailed comparisons on them in section 5.1.

We collect a large set of datasets with diverse distributions for pre-training. According to its source, we divide them into in-domain, small-scale web data, object-region data, vision data, and large-scale web data. The statistics and details are shown in Table 10. Most of them are naturally image-text pairs, while to enrich our corpus, we leverage object descriptions, region descriptions, and vision data (i.e., ImageNet). For objects and regions, we crop them from the original image according to their bounding box. The text part is composed according to a human-written template and objects. For example, the prompt template is "This image contains [OBJ_A] and [OBJ_B]", where [OBJ_A] and [OBJ_B] are two object names from the data. For vision data, because they are usually labeled with a single word or short phrase, we compose a description with prompt templates such as "A picture of [LABEL]" or "The image contains [LABEL]". For example, "A picture of cat" or "The image contains cat". We curated a dataset containing about 205.6M image-text pairs, which are available publicly on the internet. The data distribution is similar to LAION-400M. Because both are from web images, we merge them into large-scale web data (LWD).

## A.4 REPRODUCTION OF SIMVLM

Since SimVLM is not open-sourced, we need to reproduce it by ourselves. There are two main difficulties in the reproduction: 1. it uses 1.8 billion in-house data 2. the configurations (e.g., parameter size, number of layers) of its $base$ model are not clearly stated. However, there are still some clues in Section 4.4 of the SimVLM paper, where they propose a SimVLM$_{small}$ model with 8

layers, 512 embedding dimensions, and trained on about 200M web data. To demonstrate the success of our replication, we train a SimVLM$_{small}$ model with the exact same configurations on about 200M web data. We obtain a VQA score of 68.50, surpassing the reported score of 67.43 in the original paper. We argue this result verifies our successful replication.

## A.5 Effects of Compute

Our model is trained with large compute. To reveal the effects of compute, we visualize the performance improvement trends of SimVLM and DaVinci as a function of the compute spent. There are two goals: 1) to compare better with prior work, as well as to 2) to show if that level of pre-training compute was necessary. We conduct experiments on the image-to-text generation task under both zero-shot and fine-tuning settings. The results are shown in Figure 2. It is observed that with the increase in compute, both models are improved significantly and converged at 40% of compute (zero-shot), and 80% of compute (fine-tuning), respectively. Large compute is especially helpful for fine-tuning settings. After convergence, our model outperforms SimVLM consistently in these two settings.

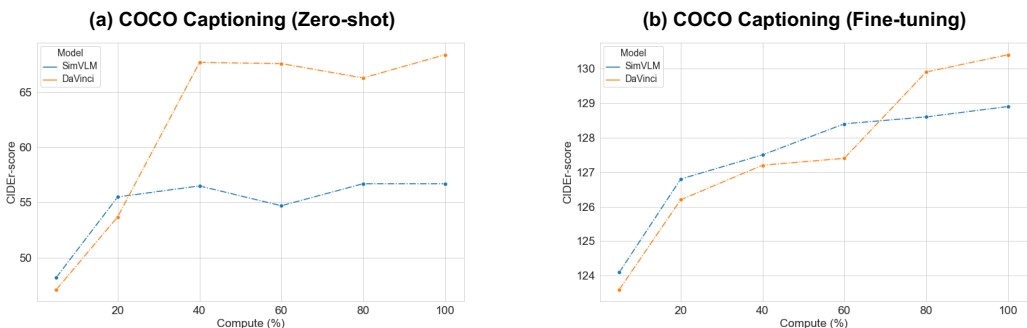

**Figure 2: The effects of compute.** X-axis is the percentage of compute and Y-axis is the CIDEr score on COCO captioning task.

## A.6 Effects of Masking Strategies

In our experiments, we adopt dynamic masking, where the masking ratio is sampled from a uniform distribution U(0, 1). The prefix ratio could be 0, where the prefix image is none, and the model is forced to predict the whole image with the input caption. There are other designs to mask images. Here we compared three different masking strategies: 1) masked image modeling (randomly masking some patches), 2) in-painting (randomly masking some continuous spans in the middle of the image), and 3) suffix-painting (ours). The results are shown in Table 11. Both masked image modeling and in-painting are effective and competitive. It is observed that suffix-painting is better than masked image modeling and in-painting across all tasks, demonstrating that suffix-painting works well.

| Method | COCO B@4 / C | VQA Acc | SNLI-VE Acc | NLVR2 Acc | ImageNet Acc | Food101 Acc | CIFAR10 Acc | MNLI Acc | SST-2 Acc | Text2Image IS / FID |
|---|---|---|---|---|---|---|---|---|---|---|
| No Pre-training | 32.1 / 96.71 | 52.73 | 54.23 | 51.08 | –* | –* | –* | 66.32 | 79.84 | –* |
| MIM | 34.7 / 113.4 | 68.18 | 75.34 | 69.66 | 48.46 | 56.95 | 72.79 | 81.72 | 89.84 | 9.50 / 74.13 |
| In-painting | 34.5 / 112.5 | 67.46 | 75.41 | 68.66 | 47.50 | 54.38 | 71.20 | 81.55 | 89.84 | 9.97 / 68.15 |
| Suffix-painting (ours) | **35.8 / 117.3** | **69.25** | **76.22** | 72.55 | **48.88** | **75.32** | **73.82** | **81.76** | **90.25** | **12.35 / 53.14** |
| Token Projection | 17.7 / 49.2 | 52.13 | 71.11 | 52.01 | 15.11 | 25.62 | 61.01 | 82.01 | 90.25 | 11.89 / 60.96 |
| Patch Projection | 25.7 / 79.5 | 57.69 | 71.92 | 57.45 | 36.23 | 44.31 | 69.40 | 81.73 | 90.05 | 11.41 / 61.87 |
| ResNet Feature (ours) | **35.8 / 117.3** | **69.25** | **76.22** | 72.55 | **48.88** | **75.32** | **73.82** | **81.76** | **90.25** | **12.35 / 53.14** |

**Table 11: The effects of masking strategies and image feature extraction on COCO Captions, VQA, SNLI-VE, NLVR2, ImageNet, Food101, CIFAR10, MNLI, SST-2, and text-to-image generation.** MIM denotes masked image modeling, where some patches are randomly sampled and masked. Because linear probe and zero-shot text-to-image generation require a pre-trained model to be frozen, the "No Pre-training" results on ImageNet, Food101, CIFAR10, and Text2Image are not reported and labeled by *.

### A.7 EFFECTS OF IMAGE FEATURE EXTRACTION

There are several different ways to extract image features. We compare three different image representation methods: 1) token projection (projecting the prefix tokens to the hidden dimension of the backbone network on the token-level), 2) patch projection (similar to ViT embedding, we split an image into fixed-size patches, embed each of them by a trainable linear projection on the pixel-level), and 3) ResNet feature extraction (ours). The comparison is shown in Table 11. From the results, we observed that ResNet feature extraction outperforms token projection and patch projection by a large margin. Therefore, we decided to adopt ResNet to extract image features.

### A.8 SCALING EFFECTS OF DATA SIZE

In this section, we explore the scaling effects of our model. We plot the trends with the increase in data size on four tasks: COCO captioning, VQA, SNLI-VE, and NLVR2. The performance improvement shown in Figure 3 demonstrates that both SimVLM and DAVINCI are scaling well with pre-training data size. In addition, DAVINCI consistently outperforms SimVLM on different data sizes across these tasks.

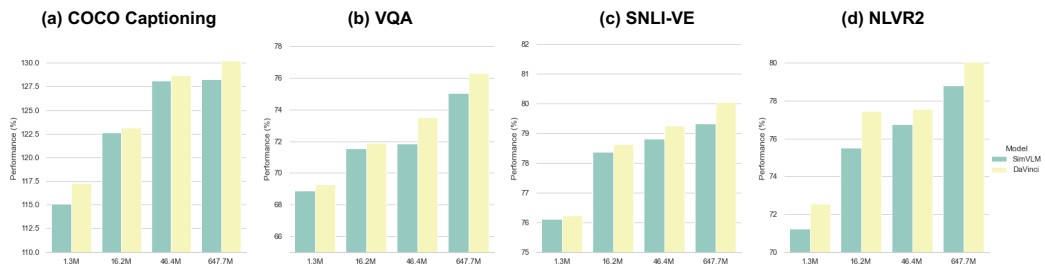

**Figure 3: The scaling effects of data size.**

### A.9 FULL COMPARISON WITH EXISTING METHODS

In Table 12, we display a comprehensive comparison with state-of-the-art vision-language models on vision, language, and multi-modal downstream tasks.

### A.10 LIMITATION AND SOCIETAL IMPACTS

**Limitation.** Like most of the previous pre-training studies, the entire project consumed 40 V100 GPU years on an in-house computing cluster with large electricity costs. We tried to keep our model size small enough, but there is still potential for efficiency improvements such as sparse training (Zhou et al., 2021d;c), dataset distillation (Zhou et al., 2022c), and progressive training (Rusu et al., 2016). We will explore those techniques to improve the training efficiency and reduce the carbon footprint so that it can adhere to proposals on "green" deep learning (Schwartz et al., 2020; Xu et al., 2021b). Furthermore, although we have tried our best to include as many tasks as we can to demonstrate the versatility of DAVINCI, we believe our method can be expanded to more tasks (e.g., machine translation, summarization, object detection, etc.), modalities (e.g., video and speech). We leave these investigations to future work.

**Potential Societal Impacts.** Our model has image generation ability with risk of abuse, like fake portraits on social media (Hill & White, 2020), which is a common potential risk in image generation research. Viable solutions are watermarking (Yu et al., 2021) and introducing a strict user license.

### A.11 VISUALIZATION OF IMAGE GENERATION

In this section, we conduct a qualitative analysis by visualizing the generation samples. Figure 4 shows the comparison with DALLE and OFA with the same query. More generated samples are shown in Figures 5.

| Model | #Params. | Text MNLI Acc | Vision ImageNet LE / FT | Image2Text COCO B@4 / C | Text2Image COCO IS↑ / FID↓ | Multi-modal VQA test-dev / test-std | Multi-modal NLVR2 dev / test-P |
|---|---|---|---|---|---|---|---|
| *Encoder-only Multi-modal Models* | | | | | | | |
| VisualBERT (Li et al., 2019) | 170M | 81.60 | – | – | – | 70.80 / 71.00 | 67.40 / 67.00 |
| ViLBERT (Lu et al., 2019) | 274M | 79.90 | – | – | – | 70.55 / 70.92 | – |
| VL-BERT (Su et al., 2020) | 170M | 81.20 | – | – | – | 71.16 / – | – |
| LXMERT (Tan & Bansal, 2019a) | 240M | 80.40 | – | – | – | 72.42 / 72.54 | 74.90 / 74.50 |
| OSCAR (Li et al., 2020) | 155M | – | – | 36.5 / 123.7 | – | 73.16 / 73.44 | 78.07 / 78.36 |
| VinVL (Zhang et al., 2021b) | 157M | – | – | 38.2 / 129.3 | – | 75.95 / 76.12 | 82.05 / 83.08 |
| ViLT (Kim et al., 2021) | 88M | – | – | – | – | 70.85 / – | 74.91 / 75.57 |
| ALBEF (Li et al., 2021) | 210M | – | – | – | – | 75.84 / 76.04 | 82.55 / 83.14 |
| X-VLM (Zeng et al., 2021) | 240M | – | – | 39.6 / 132.6 | – | 78.22 / 78.37 | 84.41 / 84.76 |
| VLMO (Wang et al., 2021a) | – | – | – | – | – | 76.64 / 76.89 | 82.77 / 83.34 |
| *Encoder-Decoder Multi-modal Models* | | | | | | | |
| UNICORN (Yang et al., 2021) | – | – | – | 35.8 / 119.1 | – | 69.20 / 69.40 | – / – |
| Uni-ENDN (Li et al., 2022b) | 110M | – | – | – | – | 72.20 / 72.50 | – / – |
| Pixel-BERT (Huang et al., 2020) | 144M | – | – | – | – | 74.45 / 74.55 | 76.50 / 77.20 |
| E2E-VLP (Xu et al., 2021a) | 94M | – | – | 36.2 / 117.3 | – | 73.25 / 73.67 | 77.25 / 77.96 |
| VL-T5 (Cho et al., 2021) | 220M | – | – | 34.5 / 116.5 | – | – / 70.30 | 74.60 / 73.60 |
| VL-BART (Cho et al., 2021) | 220M | – | – | 35.1 / 116.6 | – | – / 71.30 | 71.70 / 70.30 |
| *Text2Image Models* | | | | | | | |
| AttnGAN (Xu et al., 2018) | | – | – | – | 23.30 / 35.20 | – / – | – / – |
| DM-GAN (Zhu et al., 2019) | | – | – | – | 32.20 / 26.50 | – / – | – / – |
| DALLE (Ramesh et al., 2021) (250M) | 12B | – | – | – | 17.90 / 27.50 | – / – | – / – |
| DALLE (Ramesh et al., 2021) (640M)[†] | 82M | – | – | – | 15.79 / 29.22 | – / – | – / – |
| CogView (Ding et al., 2021) | 4B | – | – | – | 18.20 / 27.10 | – / – | – / – |
| *Unified Models* | | | | | | | |
| Unifying (Huang et al., 2021) | 228M | – | – | 37.3 / 122.6 | – / 29.90 | – / – | – / – |
| FLAVA (Singh et al., 2021) | 240M | 80.33 | 75.54 / – | – | – | 72.80 / 72.49 | – / – |
| SimVLM (Wang et al., 2021b) (640M)[†] | 153M | **83.27** | 76.04 / – | 38.5 / 128.7 | – | 75.04 / 75.03 | 78.82 / 79.72 |
| SimVLM (Wang et al., 2021b) (1.8B) | | 83.40 | 80.60 / – | 39.0 / 134.8 | – | 77.87 / 78.14 | 81.72 / 81.77 |
| OFA (Wang et al., 2022) | 182M | 84.30 | – / 82.20 | 41.0 / 138.2 | 21.50* / 20.80* | 78.00 / 78.10 | – / – |
| Florence (Yuan et al., 2021) | 637M | – | – / 90.05 | – / – | – / – | 80.16 / 80.36 | – / – |
| DAVINCI | 154M | 83.13 | **78.81 / 83.92** | 39.2 / 130.4 | **17.44 (22.41*) / 24.21 (19.82*)** | 76.32 / 76.44 | **80.03 / 80.25** |

**Table 12: Comparison with state-of-the-art vision-language models on vision, language, and multi-modal downstream tasks.** All results are from *base*-size models. LE and FT denote linear evaluation and fine-tuning performance, respectively. Image2Text results are reported without CIDEr optimization. [†] are our reproduced models. [*] are the results after fine-tuning. SimVLM (1.8B) and OFA are pre-trained with much larger corpus or human-labeled data of many downstream tasks, and thus they are not comparable and are labeled in gray. Florence (Yuan et al., 2021) is pre-trained with a much larger model size (Florence-CoSwin-H, 637M) and more pre-training data (900M), so the numbers are in grey. **bold** denotes the best across unified models.

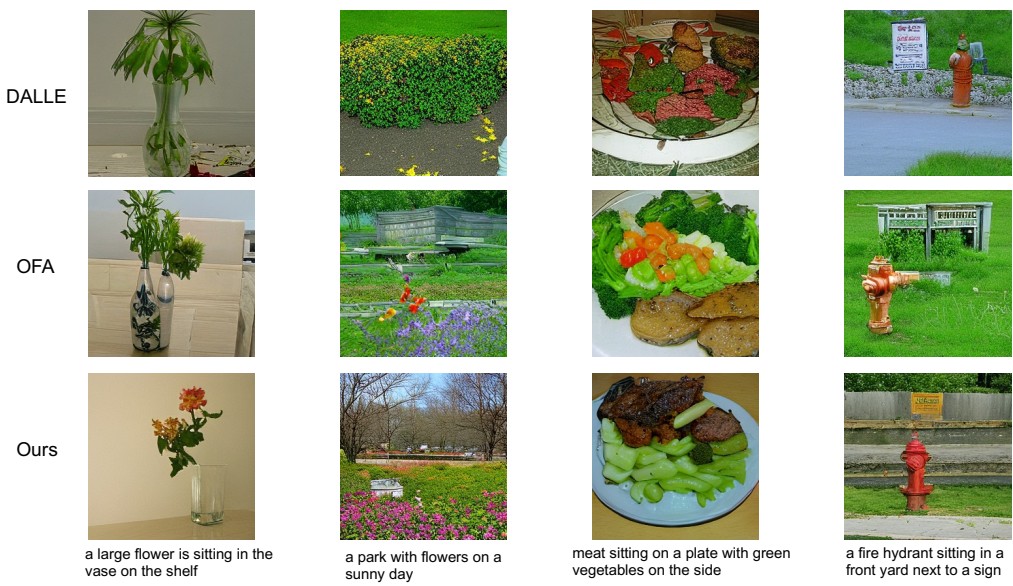

**Figure 4:** Comparison with DALLE and OFA on text-to-image generation.

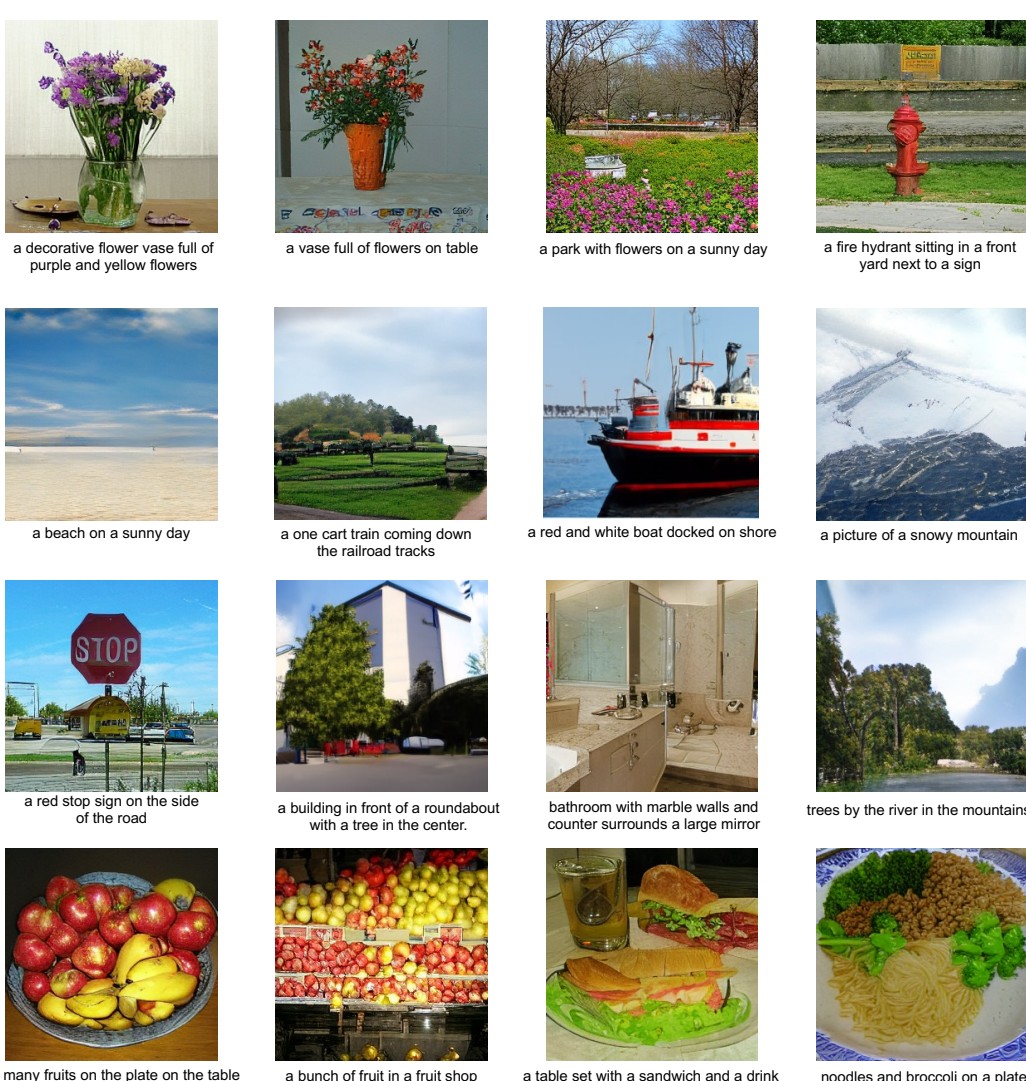

**Figure 5:** Generation samples by DAVINCI.

