# OpenReview forum: "Write and Paint: Generative Vision-Language Models are Unified Modal Learners"
_ICLR.cc/2023/Conference — ICLR 2023 poster_

### Official Review · Reviewer_wium · 2022-10-14

**Confidence:** 3
**Correctness:** 3
**Technical Novelty And Significance:** 2
**Empirical Novelty And Significance:** 3
**Recommendation:** 8

**Clarity, Quality, Novelty And Reproducibility:**

The paper seems clear and well-written. I think the idea is mostly novel, but with significant overlap to CM3 from what I understand, which I'd like the authors to clarify in rebuttal.

**Strength And Weaknesses:**

Strengths:
* The proposed model does well while being quite simple, which in turn suggests it has potential to be widely adopted (especially considering how many transformer implementations are already very well-optimized). The paper is easy to read as well.
* I appreciate the comparison with a variety of past models covering NLP, Image Classification, and vision-and-language.

Weaknesses:
* I'd like to see a comparison with this paper if possible CM3 A Causal Masked Multimodal Model of the Internet https://arxiv.org/abs/2201.07520 - because it (like this proposed approach) uses a Discrete VAE token-based approach.
* Most of the tasks specified in the vision-and-language space seem focused on literal recognition of what's in images. I'd be curious as to how this model stacks up on Visual Commonsense Reasoning (visualcommonsense.com) for instance.

Perhaps less important but still important: I'm a fan of this paper overall, but I do wonder why it takes 10 days to train on 1024 V100 GPU's. That's quite a lot for a BERT-Base model -- and in turn, it makes me think that one of the main reasons this paper does so well is because it was just trained with more effective compute than FLAVA or SimVLM. That doesn't mean that this paper is bad by any means, but I would be happy to increase my score if there was some experiment showing e.g. the performance of DaVinci over the course of pretraining -- e.g. as a function of the compute spent, both a) to compare better with prior work, as well as to b) show if that level of pretraining compute was necessary.

Another related question relates to the ablations in Table 5. For PLM only vs. PIM only, vs. the full model -- are the ablations only trained with half the effective compute as the full model? (as they're trained on half as many sequences) I'd be curious as to how they do given double the pretraining compute.

**Summary Of The Paper:**

This paper proposes a vision-and-language transformer that is multi-tasked to both generate an image completion, given language ("Prefix Image Modeling") and to both generate language given images ("Prefix Language Modeling"). The key claim of this paper is that by multitasking these two settings, the resulting model is able to do well on a wider range of vision-and-language tasks.

Despite not being a very large model, (it's BERT-Base sized), it performs favorably to SimVLM and FLAVA (two recent very large-scale approaches) in a finetuned setting, on both language-only and vision-language tasks. An ablation study (Table 5) suggests that these objectives are complementary.

**Summary Of The Review:**

I'm leaning positive on this paper but I'd like some of my concerns to be addressed in the author response first :)

---

Update post-review: I think the authors addressed my main concerns, so I'm bumping up my score to 8.

---

> ### Author Response · Authors · 2022-11-11
> **Response to Reviewer wium (1/2)**
>
> Dear Reviewer wium,
>
> Thank you very much for your comprehensive review and valuable feedback! We address your comments one by one as follows:
>
> **[CM3]**
>
> CM3 is a causal masked multi-modal model learned from web data. There are several key differences between CM3 and our model.
>
> [Data] We are using image-text pairs following previous vision-language pre-training studies (e.g., Oscar, SimVLM), while CM3 does not rely on paired data. Both methods have their own advantages: image-text pairs introduce important image-text alignment information to DaVinci, and unpaired data makes CM3 easily leverage more internet data.
>
> [Model Architecture] CM3 uses a decoder-only model while we adopt encoder-decoder model architecture.
>
> [Objectives] CM3 proposes a causally masked objective that can do causal language modeling while optionally allowing for bidirectionality when needed. In contrast, we propose symmetric prefix image-language objectives to better exploit the self-supervision from paired image-text data. Specifically, our proposed objective encourages the model to learn to generate (or reconstruct) data in one modality based on the prefix in the same modality and the complete input in the other modality. In doing so, our model can better capture vision-language alignment.
>
> [Experiments & Target Tasks] CM3 focuses on the zero-shot and few-shot image / text generation. Our work conducts 27 tasks spanning vision-only, text-only, and multi-modal understanding and generation. Because DaVinci leverages image-text pairs with visual-language alignment information, it is good at multi-modal understanding tasks (e.g., VQA, VE, NLVR), which are not included in the CM3 paper. Multi-modal understanding tasks require the vision-language alignment ability, and models learned from image-text pairs have the advantage over that learned from unpaired data.
>
> Despite the different experimental designs, DaVinci has a common task with CM3: zero-shot image generation. On this task, we achieve better performance (24.21) than CM3-Large (29.56). Note that CM3-Large is a 13B model and much larger than our model. Another common task is image captioning. However, it is not directly comparable, because they use different evaluation metrics.
>
> In sum, a key difference in terms of the pre-training objective is that we are doing encoder-decoder conditional generation while CM3 is doing causal language modeling (with a new masked strategy). The difference between pre-training objectives in our work and CM3 originates from the difference in data sources, resulting in different targeted downstream tasks. Each model design has its own merits.
>
> We have discussed and cited CM3 in our revised version. Thanks for your question!
>
>
> **[Visual Commonsense Reasoning]**
>
> Thanks for your consideration of the generalization to more diverse tasks!
> Our model is an encoder-decoder framework and can be generalized to Visual Commonsense Reasoning following the idea of multi-choice fine-tuning. The process is similar to NLVR2 (Natural Language for Visual Reasoning), which is also a reasoning task conducted in our experiments. Commonsense reasoning is an under-explored task in vision-language pre-training, but it is essential and meaningful. We will systematically explore commonsense reasoning abilities in our future work. Thanks very much for your suggestion!

---

> > ### Author Response · Authors · 2022-11-11
> > **Response to Reviewer wium (2/2)**
> >
> > **[A Function of the Compute]**
> >
> > Thanks for your question!
> > The model trained on the largest dataset takes lots of computational resources because: 1) the dataset is large, containing about 640M image-text pairs 2) the sequence length is long, which is 512 in our experiments. As we know, BERT is trained with a sequence length of 128 for the first 90% of the steps and then trained with a sequence length of 512 for the rest 10% of the steps to speed up pre-training. However, we follow RoBERTa to train DaVinci with a sequence length of 512 for all steps to obtain a better performance. Because attention is quadratic to the sequence length, longer sequences are disproportionately expensive. In addition, the vision sequence length is 256 because a 256*256 image input tokenized by the VQGAN tokenizer is 256.
> > We strongly agree that revealing the effects of computational resources is helpful in understanding the pre-training behavior, so we conduct more experiments and visualize the relation between performance and computational resources. We have updated these results in our revised paper (Appendix A.7 and Figure 3). It is observed that with the increase in compute, both models are improved significantly and converged at 40\% of compute (zero-shot), and 80\% of compute (fine-tuning), respectively. Large compute is especially helpful for fine-tuning. After convergence, our model outperforms SimVLM consistently in these two settings.
> > In addition to the scaling effects of compute, we added more analyses about the scaling effects of pre-training data. The newly added Appendix A.10 and figure 4 demonstrate the good scalability of our model with the increase of pre-training data.
> >
> > **[PIM-only and PLM-only]**
> >
> > We agree that the compute may affect performance. Therefore, we updated Table 5 with new results of our model with half compute to align their compute. It is shown our observations and discussions in Section 5.2 still hold. Thanks for your question! We have included these results in our revised version following your valuable suggestion.
> >
> > | Model \ Dataset  | COCO | VQA | VE | NLVR2 | ImageNet | Food101 | CIFAR10 | MNLI | SST-2 | Text2Image (IS/FID)
> > | ----------- | ----------- | ----------- | ----------- | ----------- | ----------- | ----------- | ----------- | ----------- | ----------- | ----------- |
> > | Ours-PLM | 33.6 / 111.17 | 65.15 | 73.91 | 53.28 | 48.05 | 74.17 | 72.98 | 81.42 | 89.97 | 10.26/59.64 |
> > | Ours-PIM | 34.3 / 116.58 | 68.89 | 75.79 | 69.78 | 45.54 | 71.18 | 70.11 | 81.94 | 90.53 | -- |
> > | Ours (half) | 35.7 / 117.15 | 68.93 | 76.06 | 72.45 | 48.27 | 74.96 | 73.62 | 81.52 | 90.11 | 12.06/55.83
> > | Ours | 35.8 / 117.30 | 69.25 | 76.22 | 72.55 | 48.88 | 75.32 | 73.82 | 81.76 | 90.25 | 12.35/53.14 |

---

> > > ### Comment · Reviewer_wium · 2022-11-24
> > > **thanks for the response!**
> > >
> > > Thanks for the response! I think this addresses my concerns, so I'll bump up my score from 6 -> 8. I think this is a result that the community ought to be aware of. I think it's interesting how this approach seems to outperform flava and simvlm; I think the ablations help but I'm not 100% sure I understand the takeaway message from them.

---

> > > > ### Author Response · Authors · 2022-11-24
> > > > **Thanks for raising the rating score**
> > > >
> > > > Dear Reviewer wium,
> > > >
> > > > We would like to thank you again for your effort and positive feedback!
> > > >
> > > > We are very happy that our response and updated presentation have addressed your concerns. Following your great suggestions, we added more experiments on compute to understand and reveal the source of performance gains. We are very grateful for your constructive and valuable comments, which helped improve our manuscript a lot and made the paper stronger. We hope our responses have answered your questions and we are happy to answer any questions you may have later.
> > > >
> > > > Thank you very much!

---

### Official Review · Reviewer_qYLj · 2022-10-25

**Confidence:** 4
**Correctness:** 3
**Technical Novelty And Significance:** 3
**Empirical Novelty And Significance:** 3
**Recommendation:** 6

**Clarity, Quality, Novelty And Reproducibility:**

Most of the part of the paper is clear, however, some of the details of the experiment and model setting is missing,

**Strength And Weaknesses:**

[Strength]

- The proposed method combines prefix language modeling with prefix image modeling, which is a natural and simple extension.

-  Experiments ablating different objectives, as well as different dataset sources and showing the effectiveness of the different types of pre-training data.

- Similar to OFA, the proposed method can do text-to-image generation tasks as well as vision and language understanding tasks.

[Weakness]

- From the introduction, the proposed method is motivated by auto-regressive models like GPT3 and PaLM. GPT3 has few shot abilities, and PaLM can multi-task and even do zero-shot tasks. One key property of these two models is they can use the same set of parameters for many tasks. However, the proposed method still falls into the first pretrain and then finetune formula.

- The authors claim that VL-T5 and OFA are hard to scale up since it is non-trivial to collect a large number of vision and language datasets for pre-training. I hold a different opinion, the pre-training tasks (mask language modeling and mask image modeling) can be one of the tasks, and thus the model still benefits from a large training corpus.

- Another big concern of the proposed method is the model size. The paper only shows models with 152M parameters. Thus it is not clear whether the proposed method can scale to a large model.

- The pre-training datasets also contain supervised datasets such as COCO, refer coco, VG, etc. Given the claim of OFA and VL-T5, the proposed method suffers a similar scaling problem.

- In Table 3, the author didn't compare OFA and simVLM (1.8B). Although OFA uses some additional tasks, its pre-training corpus is substantially smaller compared to the proposed approach.

- for prefix language modeling, the author randomly samples the mask ratio,  is it from (0, 1)? or some other span? Table 5 shows the results of the ablation study. I am more interested in the effect of image masking strategies: such as random masking vs. masking at the end. Different ratios of masking etc. However, there is no experiment on that.

**Summary Of The Paper:**

This paper proposed a new unified model named Davinci, which is trained with prefix language modeling and prefix image modeling. Davinci is able to do images and text generation as well as other vision and language understanding tasks. The authors carefully benchmark the performance of different vision and language pre-training objectives on different scales of pre-training datasets and achieve competitive performance.

**Summary Of The Review:**

This paper proposed a new unified model named Davinci, which is trained with prefix language modeling and prefix image modeling. The proposed method is simple and shows some potential for training a unified model. However, there is multiple weakness in the paper that require the authors furhur clarify. (check the weakness section for details).

---

> ### Author Response · Authors · 2022-11-11
> **Response to Reviewer qYLj (1/2)**
>
> Dear Reviewer qYLj,
>
> Thank you very much for your comprehensive review and valuable feedback! We address your comments one by one as follows:
>
> **[Zero-shot and Few-shot Tasks]**
>
> Zero-shot and few-shot abilities are two essential abilities of the generative model. In our submission, we conducted experiments to verify our model's abilities on zero-shot and few-shot tasks. The results were shown in Table 2, and there was a separate paragraph discussing zero-shot and few-shot transfer in Section 4.4. DaVinci demonstrates good generalization ability on VQA, VE, NLVR2, image-to-text, text-to-image, and NoCaps. In addition to these datasets, we conduct experiments on a recently proposed benchmark, VLUE zero-shot captioning. The superior performance on VLUE further reveals that our model is able to generalize to out-of-domain datasets rather than limiting to COCO domain.
> Our model size is only about 150M, and it has demonstrated good zero-shot and few-shot abilities. Considering that scaling to super-large models like GPT-3 and PaLM will bring better zero-shot capabilities, we leave the exploration for future work.
>
> **[VL-T5 and OFA]**
>
> VL-T5, OFA and DaVinci adopt two different technical routes: VL-T5 and OFA fall into the multi-task learning paradigm, while DaVinci falls into the self-supervised pre-training paradigm. Each has its own merits. They are not directly comparable because self-supervised learning does not rely on the human-annotated labels during pre-training, while multi-task learning requires them. The multi-task learning model depends on human-annotated data for each task, leading to the fast acquisition of task-specific knowledge. Our self-supervised pre-training does not rely on human-annotated data (especially the labels), leading to better scalability and versatility. Many previous works under the self-supervised pre-training paradigm (e.g., SimVLM) take lots of pre-training data to achieve good performance, and works under the multi-task learning paradigm (e.g., VL-T5) adopt fewer but human-annotated data.
> We conducted comparisons with OFA and found that our model has better image recognition and generation abilities. The results are shown as follows. For linear probe tasks, our model outperforms OFA by a large margin, with an average of 9.02% improvement. For the text-to-image generation task, our model outperforms OFA after fine-tuning (FID 19.82 (ours) v.s. 20.80 (OFA), the lower the better). Note that OFA did not conduct zero-shot text-to-image generation in their paper.
>
> | Model \ Dataset  | ImageNet1K(fine-tuning) | ImageNet1K | Food101 | CIFAR10 | CIFAR100 | Cars | Aircraft | DTD | Pets | Flowers102 | MNIST | STL10 | Country211 | Text2Image (IS/FID)
> | ----------- | ----------- | ----------- | ----------- | ----------- | ----------- | ----------- | ----------- | ----------- | ----------- | ----------- | ----------- | ----------- | ----------- | ----------- |
> | OFA-Base | 82.2 | 71.4 | 75.2 | 86.1 | 66.7 | 54.9 | 30.9 | 70.3 | 81.0 | 86.3 | 97.4 | 96.4 | 17.9 | 21.50/20.80 |
> | DaVinci | 83.9 | 77.7 | 90.1 | 94.0 | 80.1 | 74.6 | 49.6 | 78.3 | 88.2 | 96.9 | 99.0 | 99.2 | 29.9 | 22.41/19.82 |
>
> **[Model Size]**
>
> Thanks for your question!
> Due to the limit of computational resources, we only conducted experiments with base model size.
> To verify our model's scalability, we added new experiments with DaVinci_large model, which achieves consistent improvement over image-to-text / text-to-image generation, multi-modal understanding, and vision / language understanding tasks. It demonstrates our model's good scalability in single-modal and multi-modal understanding and generation with the increase in model size.
> Due to the time limitation, we only train the large model on the in-domain dataset (1.3M image-text pairs). We will continue to explore larger model sizes trained on larger datasets, which is an important future research direction for us. Thanks for your question!
>
>
> | Model \ Dataset  | COCO Captioning | VQA | VE | NLVR2 | ImageNet | CIFAR10 | MNLI | SST-2 | Text2Image (IS) | Average | Text2Image (FID) |
> | ----------- | ----------- | ----------- | ----------- | ----------- | ----------- | ----------- | ----------- | ----------- | ----------- | ----------- | ----------- |
> | DaVinci_base | 117.3 | 69.25 | 76.22 | 72.55 | 48.88 | 73.82 | 81.76 | 90.25 | 12.35 | 71.38 | 53.14 |
> | DaVinci_large | 118.8 | 69.83 | 76.83 | 73.16 | 58.45 | 80.23 | 83.61 | 91.13 | 13.02 | 73.90 | 40.18 |

---

> > ### Author Response · Authors · 2022-11-11
> > **Response to Reviewer qYLj (2/2)**
> >
> > **[Pre-training Data]**
> >
> > The method that we use labeled data is fundamentally different from OFA/VL-T5. Our model design is a self-supervised method and does not rely on human-annotated labels and supervised objectives / losses. For example, OFA requires the object positions and object names for object detection and both questions and answers for VQA during pre-training. Given the refcoco dataset, OFA learns to generate a sequence of object positions and object names while our model simply constructs image-text pairs from images and object names. Although we explored the effectiveness of supervised datasets, our model works well without supervised datasets, which has been carefully verified in our ablation study (Table 4). This is a benefit of self-supervised learning over multi-task learning that relies on labeled data. We updated our manuscript to make this claim clearer.
> > In addition, as for the scalability of our model, we verify it in Section 5.1. Settings 1 and 6 do not use any supervised data. As the results shown in Table 4, the performance is improved significantly from setting 1 to setting 6, demonstrating the scalability of our model with the increase in pre-training data.
> >
> > **[Masking Strategies]**
> >
> > As for the masking ratio, we randomly sample it from a uniform distribution U(0, 1).
> > For the effects of different ratios, the results are shown in Table 5.
> > As for the masking strategies, we added new experiments to explore their effects. Here we compared three different masking strategies: 1) masked image modeling (randomly masking some patches), 2) in-painting (randomly masking some continuous spans in the middle of the image), and 3) suffix-painting (ours).
> > The results are as follows. Both masked image modeling and in-painting are effective and competitive. It is observed that suffix-painting is better than masked image modeling and in-painting across all tasks, demonstrating that suffix-painting works well.
> >
> > | Model \ Dataset  | COCO | VQA | VE | NLVR2 | ImageNet | CIFAR10 | MNLI | SST-2 | Text2Image (IS/FID)|
> > | ----------- | ----------- | ----------- | ----------- | ----------- | ----------- | ----------- | ----------- | ----------- | ----------- |
> > | masked image modeling | 34.7 / 113.4 | 68.18 | 75.34 | 69.66 | 48.46 | 72.79 | 81.72 | 89.84 | 9.50/74.13 |
> > | in-painting | 34.5 / 112.5 | 67.46 | 75.41 | 68.66 | 47.50 | 71.20 | 81.55 | 89.84 | 9.97/68.15 |
> > | suffix-painting (ours) | 35.8 / 117.3 | 69.25 | 76.22 | 72.55 | 48.88 | 73.82 | 81.76 | 90.25 | 12.35/53.14 |
> >
> > We have updated our paper with these new results in Appendix A.8, to reveal the effects of different painting strategies clearly. Thanks very much for your constructive suggestion!
> >
> >
> > Thanks for your questions! We have included these comparisons in our revised version.

---

> > > ### Comment · Reviewer_qYLj · 2022-11-23
> > > **Thanks for the responses**
> > >
> > > Thanks for the detailed response, and it mostly solves my concerns. I will update the score for this paper.

---

> > > > ### Author Response · Authors · 2022-11-23
> > > > **Thanks for raising the rating score**
> > > >
> > > > Dear Reviewer qYLj,
> > > >
> > > > We would like to thank you again for your effort and positive feedback!
> > > >
> > > > We are very happy that our response and updated presentation have resolved your concerns. We are also very grateful for your constructive and valuable comments, which helped improve our manuscript a lot and made the paper stronger.
> > > >
> > > > We really enjoyed the discussion with you. Thank you very much!

---

### Official Review · Reviewer_9zCh · 2022-10-26

**Confidence:** 3
**Correctness:** 3
**Technical Novelty And Significance:** 3
**Empirical Novelty And Significance:** 3
**Recommendation:** 6

**Clarity, Quality, Novelty And Reproducibility:**

Clarity:

The presentation of this paper is excellent.

Quality:

The paper has a good quality in general.

Novelty:

The novelty of this proposed model is moderate.

Reproducibility:

I believe this work can be reproduced easily.


**Strength And Weaknesses:**

Strengths:

1.A simple and unified foundation model for vision-language pre-training is proposed.
2.Compared with previous unified vision-language foundation models and other vision-language models, the proposed model shows highly competitive results.
3.Various ablation studies have confirmed the advantages of the proposed model.

Weaknesses:

1.The authors should conduct more rigorous experiments and comparisons. For example, in Tab. 3, the authors only listed the results of SimVLM (1.8B), OFA and Florence, and claimed that these results should not be directly compared. However, to clearly demonstrate the effectiveness of the proposed method, the authors could change the training settings (e.g., train on the same data of OFA or use a model with the same size as Florence) to make fair comparisons with these models.

**Summary Of The Paper:**

In this work, a generative vision-language model that can perform both image-to-text generation and painting text-to-image generation, called DAVINCI, is proposed. In DAVINCI, prefix language modeling and prefix image modeling are adopted as the training objective, which make the model simple and scalable. On 27 generation/understanding tasks, DAVINCI achieves highly competitive results.

**Summary Of The Review:**

This work is overall a good contribution to the research community. I would recommend acceptance of this paper.

---

> ### Author Response · Authors · 2022-11-11
> **Response to Reviewer 9zCh**
>
> Dear Reviewer 9zCh,
>
> Thank you very much for your comprehensive review and valuable feedback! We address your comments one by one as follows:
>
> **[OFA]**
>
> OFA and DaVinci adopt two different technical routes: OFA falls into the multi-task learning paradigm, while ours falls into the self-supervised pre-training paradigm. Each has its own merits. They are not directly comparable because self-supervised learning does not rely on the human-annotated labels during pre-training, while multi-task learning requires them. The multi-task learning model depends on human-annotated data for each task, leading to the fast acquisition of task-specific knowledge. Our self-supervised pre-training does not rely on human-annotated data (especially the labels), leading to better scalability and versatility. Many previous works under the self-supervised pre-training paradigm (e.g., SimVLM) take lots of pre-training data to achieve good performance, and works under the multi-task learning paradigm (e.g., VL-T5) adopt fewer but human-annotated data.
> We conducted comparisons with OFA and found that our model has better image recognition and generation abilities. The results are shown as follows. For linear probe tasks, our model outperforms OFA by a large margin, with an average of 9.02% improvement. For the text-to-image generation task, our model outperforms OFA after fine-tuning (FID 19.82 (ours) v.s. 20.80 (OFA), the lower the better). Note that OFA did not conduct zero-shot text-to-image generation in their paper.
>
>
> | Model \ Dataset  | ImageNet1K(fine-tuning) | ImageNet1K | Food101 | CIFAR10 | CIFAR100 | Cars | Aircraft | DTD | Pets | Flowers102 | MNIST | STL10 | Country211 | Text2Image (IS/FID)
> | ----------- | ----------- | ----------- | ----------- | ----------- | ----------- | ----------- | ----------- | ----------- | ----------- | ----------- | ----------- | ----------- | ----------- | ----------- |
> | OFA-Base | 82.2 | 71.4 | 75.2 | 86.1 | 66.7 | 54.9 | 30.9 | 70.3 | 81.0 | 86.3 | 97.4 | 96.4 | 17.9 | 21.50/20.80 |
> | DaVinci | 83.9 | 77.7 | 90.1 | 94.0 | 80.1 | 74.6 | 49.6 | 78.3 | 88.2 | 96.9 | 99.0 | 99.2 | 29.9 | 22.41/19.82 |
>
> **[Florence]**
>
> Thanks for your suggestion!
> Florence is a foundation model designed for computer vision and our model is designed for much broader tasks, including language-only, vision-only, and multi-modal generation and understanding tasks. What makes the comparison difficult is that Florence has a much larger model size (893M > ours 154M) and larger training data size (900M > ours 647M).
> More training data and larger model size consume lots of computation resources which are beyond our reach.
> To verify our model's scalability, we added new experiments with DaVinci_large model, which achieves consistent improvement over image-to-text / text-to-image generation, multi-modal understanding, and vision / language understanding tasks. It demonstrates our model's good scalability in single-modal and multi-modal understanding and generation with the increase in model size.
> Due to the time limitation, we only train the large model on the in-domain dataset (1.3M image-text pairs). We will continue to explore larger model sizes trained on larger datasets, which is an important future research direction for us. Thanks for your question!
>
> | Model \ Dataset  | COCO Captioning | VQA | VE | NLVR2 | ImageNet | CIFAR10 | MNLI | SST-2 | Text2Image (IS) | Average | Text2Image (FID) |
> | ----------- | ----------- | ----------- | ----------- | ----------- | ----------- | ----------- | ----------- | ----------- | ----------- | ----------- | ----------- |
> | DaVinci_base | 117.3 | 69.25 | 76.22 | 72.55 | 48.88 | 73.82 | 81.76 | 90.25 | 12.35 | 71.38 | 53.14 |
> | DaVinci_large | 118.8 | 69.83 | 76.83 | 73.16 | 58.45 | 80.23 | 83.61 | 91.13 | 13.02 | 73.90 | 40.18 |

---

### Official Review · Reviewer_NJt6 · 2022-11-05

**Confidence:** 4
**Correctness:** 3
**Technical Novelty And Significance:** 3
**Empirical Novelty And Significance:** 3
**Recommendation:** 8

**Clarity, Quality, Novelty And Reproducibility:**

Clarity & Reproducibility:
Certain details either aren’t clear to me, and I couldn’t find them in the paper (apologies if they were there):
 - On the PIM objective, it is not clear how the loss is propagated back (gumbel softmax?), or we’re just doing cross-entropy loss on the VQGAN tokens we need to generate?
 - Page 5, Table 1 - LWD - What is DaVinci 200M dataset?
 - How are the Object-Region Datasets used, i.e. what is the text part of the image example? How is an example formulated?
 - The authors don’t make it clear how “dynamic masking” is done, i.e. what is the probability distribution of the prefix sample? Is it uniformly randomly distributed, or something else, what are the parameters of this distribution? I don’t find this information anywhere.

Quality:
Paper is quite well written and is almost free of typos [1]

Novelty:
New way to pre-train VLMs, although this is novel, this is actually a straightforward extension of the way things are headed. Parti / DallE-1 both have image generation using VQGAN tokens and both PaLI and Parti have decoders.

Miscellaneous:
Neither Parti, not PaLI is mentioned and they seem relevant.


[1] Table 2, Page 7 “and they use the use mid-training”, extra “use” before mid-training.



**Strength And Weaknesses:**

Strengths:
 - All required ablations of the VLPs
 - All required ablations of the dataset sources
 - Comparisons with appropriate re-implementations of baselines.
 - Comparisons are over a varied class of task suites (NLU, Image Understanding, Multimodal Understanding, Image Generation, NL Generation) and actual benchmarks on 27 tasks.

Weaknesses:
 - Not necessarily a weakness: Since the “prefix” of the image is more seen than the suffix, I wonder if this leads to an inductive bias that the image will be less likely to paint the prefix of the image itself. i.e. say if the most important aspect of the image falls in the top-left block of the image. I wonder if the authors considered other ways of masking the image, by either just in-painting (i.e. “missing tiles”) instead of suffix-painting. This could be left for future work.
 - Also not necessarily a weakness: Right now images are treated in a hybrid manner, image input to the model is treated in pixel space (first three blocks of resnet extracts the feature maps), however the suffix that needs to be generated is in the token space (VQGAN tokens). One could have resorted to also taking the whole image as a sequence of VQGAN tokens, projecting a prefix of them to the hidden dimension of the backbone network and that prefix would be the input, while the rest of the suffix became the output. This could be left for future work.
 - Re: SimVLM base size, did the authors contact the SimVLM authors to get the exact specifics of the SimVLM base size and architecture. Although SimVLM_small seems to be replicated exactly, it is also possible that some bugs remain in the reimplementation and SMALL was small enough for these not to matter.


**Summary Of The Paper:**

The paper presents a new way to pre-train VLMs (vision-language-models) using both text generation and image generation as the VLPs (pre-training objectives).

They then ablate the objectives on their model (and show that each objective’s inductive bias is actually helping) and the datasets used for training (smaller more curated datasets being better in general) both their model and a reimplementation of SimVLM.


**Summary Of The Review:**

Leaning towards acceptance, since this seems a good enough empirical contribution and a “proof of existence” of image generation working as a pre-training objective to make the model stronger.

---

> ### Author Response · Authors · 2022-11-11
> **Response to Reviewer NJt6 (1/2)**
>
> Dear Reviewer NJt6,
>
> Thank you very much for your comprehensive review and valuable feedback! We address your comments one by one as follows:
>
> **[Masking Strategy]**
>
> In our experiments, we adopt dynamic masking, where the masking ratio is sampled from a uniform distribution U(0, 1). It is possible that the prefix ratio is 0, so the prefix image is none, and the model is forced to predict the whole image with the input caption. Considering pre-training is performed on a large dataset with lots of iterations, the top-left block of the image could be captured. Our experiments also provide support. From the results of zero-shot text-to-image generation in Figure 4, we can see that the quality of the top-left part is as good as that of other parts. Moreover, one of the critical advantages of prefixLM is the zero-shot ability for understanding and generation tasks.
> We agree that the in-painting strategy is another option, so we tried the in-painting strategy and compared it with our suffix-painting. Here we compared three different masking strategies: 1) masked image modeling (randomly masking some patches), 2) in-painting (randomly masking some continuous spans in the middle of the image), and 3) suffix-painting (ours).
> The results are as follows. Both masked image modeling and in-painting are effective and competitive. It is observed that suffix-painting is better than masked image modeling and in-painting across all tasks, demonstrating that suffix-painting works well.
>
> | Model \ Dataset  | COCO | VQA | VE | NLVR2 | ImageNet | CIFAR10 | MNLI | SST-2 | Text2Image (IS/FID)|
> | ----------- | ----------- | ----------- | ----------- | ----------- | ----------- | ----------- | ----------- | ----------- | ----------- |
> | masked image modeling | 34.7 / 113.4 | 68.18 | 75.34 | 69.66 | 48.46 | 72.79 | 81.72 | 89.84 | 9.50/74.13 |
> | in-painting | 34.5 / 112.5 | 67.46 | 75.41 | 68.66 | 47.50 | 71.20 | 81.55 | 89.84 | 9.97/68.15 |
> | suffix-painting (ours) | 35.8 / 117.3 | 69.25 | 76.22 | 72.55 | 48.88 | 73.82 | 81.76 | 90.25 | 12.35/53.14 |
>
> We have updated our paper with these new results in Appendix A.8, to reveal the effects of different painting strategies clearly. Thanks very much for your constructive suggestion!
>
> **[Image Token Representation]**
>
> Thanks for your question!
> In our preliminary experiments, we compared three different types of image representation: 1) token projection (as you suggested, projecting the prefix tokens to the hidden dimension of the backbone network on the token-level), 2) patch projection (similar to ViT embedding, we split an image into fixed-size patches, embed each of them by a trainable linear projection on the pixel-level), and 3) ResNet feature extraction. The comparison is shown as follows. From the results, we observed that 3) outperforms 1) and 2) by a large margin. Therefore, we decided to adopt ResNet to extract image features.
> Thank you very much for your constructive suggestion! We have included the details of these experiments in our revised paper.
>
> | Variant \ Dataset  | COCO | VQA | VE | NLVR2 | ImageNet | CIFAR10 | MNLI | SST-2 | Text2Image (IS/FID) |
> | ----------- | ----------- | ----------- | ----------- | ----------- | ----------- | ----------- | ----------- | ----------- | ----------- |
> | token projection | 17.7 / 49.2 | 52.13 | 71.11 | 52.01 | 15.11 | 61.01 | 82.01 | 90.25 | 11.89/60.96 |
> | patch projection | 25.7 / 79.5 | 57.69 | 71.92 | 57.45 | 36.23 | 69.40 | 81.73 | 90.05 | 11.41/61.87 |
> | ResNet feature extraction (ours) | 35.8 / 117.3 | 69.25 | 76.22 | 72.55 | 48.88 | 73.82 | 81.76 | 90.25 | 12.35/53.14 |
>
>
> **[SimVLM reimplementation]**
>
> Yes, we indeed contacted the authors of SimVLM about the exact specifics of the base size many months ago. However, they did not provide the details about base size (e.g., parameter size, number of layers). Therefore, we conducted the comparisons with completely the same data and settings to make a fair comparison. In addition, from the results of the small model, we observed a slightly better performance than their reported performance, illustrating that our model implementation and hyper-parameters are working well. Moreover, from the results in Table 4, both SimVLM and DaVinci are scaling well with the increase in data size, verifying their scalability and our successful replication. Last but not least, even if the base model uses a different model size, we can simply modify the specifics of both SimVLM and DaVinci to make a fair comparison as well.
>
> **[PIM Objective]**
>
> We are using cross-entropy loss on the VQGAN tokens that we need to generate. We have modified our draft to make it clear. Thanks for your question!

---

> > ### Comment · Reviewer_NJt6 · 2022-11-20
> > **Thanks for the responses!**
> >
> > Thanks again for the wonderful work and follow up experiments (especially: thanks for "[Image Token Representation]")!
> >
> > I had a few more questions:
> >
> > > It is possible that the prefix ratio is 0, so the prefix image is none, and the model is forced to predict the whole image with the input caption.
> >
> > Can we mention in the paper (or appendix or footnote), in what fraction of the cases it happens that the prefix image is none?
> >
> > One further question I had is from a (full-image, full-caption) dataset example, there are essentially two types of training examples we can get - (prefix-image, full-caption) and (full-image, prefix-caption) ... Can we mention in the paper if *both* of these are done, and if so, do they land up in the same training batch? Or we randomly select one of the types (if randomly, what ratio) whenever we encounter a dataset example (and so in the next epoch the drawing will be yet random)

---

> > > ### Author Response · Authors · 2022-11-21
> > > **Thanks for your responses!**
> > >
> > > Dear Reviewer NJt6,
> > > Thanks very much for your reply and recognition of our experiments!
> > >
> > > **[Fraction of the Non-image Cases]**
> > >
> > > Thanks for your question! Yes, we will include these important details in our paper.
> > > According to our experiments, the fraction of the cases in which the prefix image is none is around 1/16.
> > >
> > > **[Traning Batch]**
> > >
> > > In a training batch, both (prefix-image, full-caption) and (full-image, prefix-caption) are used and both PIM and PLM are done for a (full-image, full-caption) dataset example.
> > >
> > > Thanks again for your constructive comments and questions. Because we are in discussion stage 2, where we are not allowed to update our manuscript, we will definitely mention these important details in our final version.

---

> > > > ### Comment · Reviewer_NJt6 · 2022-11-22
> > > > **Thanks**
> > > >
> > > > Thank you, I'll update the score, I'm quite happy with the state of things.

---

> > > > > ### Author Response · Authors · 2022-11-23
> > > > > **Thanks for raising the rating score**
> > > > >
> > > > > Dear Reviewer NJt6,
> > > > >
> > > > > We would like to thank you again for your effort and positive feedback!
> > > > >
> > > > > We are very happy that our response and updated presentation have resolved your questions. We really enjoyed the discussion with you. Your valuable comments have improved our presentation a lot and made the manuscript more readable.
> > > > >
> > > > > Thank you very much! Really grateful!

---

> ### Author Response · Authors · 2022-11-11
> **Response to Reviewer NJt6 (2/2)**
>
> **[Large-Web-Dataset]**
>
> We curated a dataset containing about 205.6M image-text pairs which are available publicly on the internet. The data distribution is similar to LAION-400M. Because both of them are from web images, we merge them into large-scale web data (LWD).
> For the effect of such a large noisy web dataset, we carefully conducted an ablation study on it in Table 4, finding that our model is scaling well with them.
> We have added the details of this dataset into our revised version (Appendix A.6). Thanks for your question!
>
> **[Object-Region Dataset]**
>
> The object-region dataset is used to construct image-text pairs carrying object and region information. The text part is composed according to a human-written template and object names. For example, the prompt template is "This image contains [OBJ_A] and [OBJ_B]", where [OBJ_A] and [OBJ_B] are two object names from the dataset.
> For the effect of the object-region dataset, we conducted an ablation study on it in Table 4, finding that they are helpful for multi-model pre-training.
> Thanks very much for your question! We have added these details to our revised version (Appendix A.6).
>
> **[Dynamic Masking]**
>
> Yes, it is uniformly randomly distributed. We have included these details in our revised version's implementation part (Section 4.3). Thank you!
>
> **[PaLI and Parti]**
>
> Thanks very much for providing two relevant studies.
> PaLI trains a multilingual language-image model with an encoder-decoder model. It aims to explore effective scaling across tasks and languages. Parti is dedicated to text-to-image generation, which trains an autoregressive sequence-to-sequence model that generates image tokens from text tokens.
> We noticed that PaLI was released on Sep 14, 2022, while our paper was submitted to ICLR on Sep 28, 2022, but we are truly thankful for pointing them out. We have discussed and cited them in the updated version.
>
>
> We have also addressed other typos (e.g., Table 2, Page 7) in the updated manuscript. Thank you for your very constructive suggestions!

---

### Decision · Program_Chairs · 2023-01-20

**Decision:**

Accept: poster

**Justification For Why Not Higher Score:**

Seems interesting but maybe not good enough for spotlight?

**Justification For Why Not Lower Score:**

Paper is interesting and should be accepted.

**Metareview: Summary, Strengths And Weaknesses:**

Reviewers mostly liked the paper! Accept! Congrats!

Reviewers thought that
- "This work is overall a good contribution to the research community."
-" Leaning towards acceptance, since this seems a good enough empirical contribution and a “proof of existence” of image generation working as a pre-training objective to make the model stronger."

Authors did a good job in convincing the reviewer to increase their score.

**Note From Pc:**

if the above contains the word "oral" or "spotlight" please see: "oral" presentation means -> notable-top-5% and "spotlight" means -> notable-top-25%. As stated in our emails, we are disassociating presentation type from AC recommendations